# Can Vision Language Models Learn Intuitive Physics From Interaction?

## Abstract

Pre-trained vision language models do not have good intuitions about the physical world. Recent work has shown that supervised fine-tuning can improve model performance on simple physical tasks. However, fine-tuned models do not appear to learn robust physical rules that can generalize to new contexts. Based on research in cognitive science, we hypothesize that models need to interact with an environment to properly learn its physical dynamics. We train models that learn through interaction with the environment using reinforcement learning, as well as models that learn without interaction using supervised fine-tuning. While both reinforcement learning and supervised fine-tuning appear to improve within-task performance, they fail to produce models with generalizable physical intuitions. Models trained on one task do not reliably generalize to related tasks, even if they share visual statistics and physical principles, and regardless of whether they are trained through interaction.

## 1 Introduction

A central goal of machine learning research is to build machines that think and behave like people do. Lake et al. (2017) propose that human-like machine learning models must be capable of reasoning about their environment and its physical, social, and causal structure. These capabilities are often referred to as intuitive theories (Baillargeon et al., 1995; Spelke, 1990; Spelke & Kinzler, 2007). Here, we focus on *intuitive physics* — the ability to understand and predict the physical properties and interactions of objects (Battaglia et al., 2012; Piloto et al., 2022).

Recent work has established that vision language models (VLMs), models that receive visual and textual inputs, are still limited in their understanding of the physical world and its causal structure (Jin et al., 2023; Balazadeh et al., 2024). VLMs do not perform well on standard visual cognition tasks — such as tasks testing intuitive physics — and they do not show a good fit with human behavioral data (Schulze Buschoff et al., 2025a). While supervised fine-tuning enables models to perform well on the tasks they were fine-tuned on, they do not appear to learn generalizable intuitions about the physical world (Schulze Buschoff et al., 2025b).

A prominent idea in cognitive science is that humans learn a robust understanding of their world by interacting with it (Gibson, 1979; Merleau-Ponty, 1945; Varela et al., 1991). The key claim is that humans learn robust, generalizable concepts for explaining and predicting their world not merely from passive observation and symbolic abstraction, but from actively interacting with their environment's dynamics (Barsalou, 1999; Clark, 1998). Some have argued that directly experimenting with the physical properties of objects in the environment allows children to test their hypotheses about their environment (Gopnik et al., 1999). In contrast to passively observing the interactions of other people with an environment, they learn much more from trying, and often failing, to predict how the environment will evolve given their own actions (Smith, 1982; Chu & Schulz, 2020; Nicolopoulou, 1993; Smith & Gasser, 2005; Schulz & Bonawitz, 2007). While the important role of interaction is slowly being recognized in generative model training (Silver & Sutton, 2025; Motamed et al., 2025), its merit for teaching vision language models visual cognitive abilities such as intuitive physics has not yet been explored.

In this paper, we present a first attempt at evaluating the role of interaction for learning intuitive physics in VLMs. Interaction can be operationalized in several ways (Shapiro & Spaulding, 2025), from one- and multi-step reinforcement learning (RL) to multi-sensory robotics. We operationalize

interaction in the context of one-step RL, defining an *environment*, *action space*, and *reward function* (Sutton et al., 1998). VLMs are presented with an image of a stack of colored blocks generated by a physics engine. They must for example respond with an action sequence to move another block to build a taller, stable tower, receiving a reward that depends on the stability of the resulting tower.

We compare models that are trained to build towers through trial-and-error (the *interactive* condition) with models that are shown examples of optimal action sequences to build stable towers (the *non-interactive* condition). Similarly to how children appear to learn generalizable physical intuitions by playing with objects (Piaget, 1952), we propose that learning to build towers through interaction with the physics of the environment will enable VLMs to learn those same intuitions.

Following this line of argument, we hypothesize the following:

1. Models in the interactive condition will generalize better to building new towers not seen in their training data, compared to the non-interactive condition.

2. Models in the interactive condition will generalize better to a new task, such as judging the stability of a tower, compared to the non-interactive condition.

3. Given further training on a new intuitive physics task, models in the interactive condition will learn more quickly than the non-interactive condition.

We test these hypotheses mainly by evaluating the textual outputs of VLMs. However, it is possible that models might have the knowledge required to solve the task, but cannot produce textual outputs in the right format. We explore this distinction between model *competence* and model *performance* (Chomsky, 1965) by decoding model activations layer-wise to see how predictive they are of key physical quantities. We thus further hypothesize that these quantities will be more decodable at later model layers in models trained in the interactive condition compared to the non-interactive condition.

We find no noticeable differences between the interactive and non-interactive conditions, both in and outside of the training tasks. Both methods yield models that perform at ceiling on the tasks they are trained on, but neither method produces models that reliably generalize to new physical tasks. While we find that physical quantities like tower stability are highly decodable from model activations, neither post-training method successfully converts this competence into reliable performance on new tasks.

## 2 RELATED WORK

Despite recent advances in architectures and training methods, VLMs continue to struggle on simple visual tasks that are trivial for any human observer, such as counting objects in a scene or making judgements about their interactions (Rahmanzadehgervi et al., 2024; Schulze Buschoff et al., 2025a; Balazadeh et al., 2024). Campbell et al. (2024) suggest that these failures originate from the fact that the test images contain multiple objects whose higher-order relations must be tracked. There is evidence that pre-trained VLMs struggle to attend to and distinguish multiple objects at the same time (Frankland et al., 2021).

Supervised fine-tuning (SFT) has emerged as an efficient way to overcome limitations such as these through extensive post-training on specific problems (Han et al., 2024). These methods have also proven useful for aligning models towards more human-like outputs (Binz et al., 2024; Hussain et al., 2024). However, SFT with VLMs appears to have a limited effect on their ability to learn generalizable physical intuitions (Schulze Buschoff et al., 2025b) and interact reliably with physical environments (Mecattaf et al., 2024). One plausible hypothesis is that SFT simply allows VLMs to learn useful shortcuts for specific tasks (Geirhos et al., 2020). Indeed, Motamed et al. (2025) argue that large generative video models are likely making predictions about the physical world without a proper understanding of its underlying physics. They suggest that a lack of active interaction with the physical world could be the limiting factor. Our study therefore seeks to explore whether models' understanding of physics can be improved through active interaction with an environment.

In line with this proposal, online reinforcement learning (RL), a paradigm in which models learn through interaction with an environment, has recently been argued to generalize more robustly than SFT (Chu et al., 2025), an analogue of offline RL (Wu et al., 2025). Online RL refers to updating

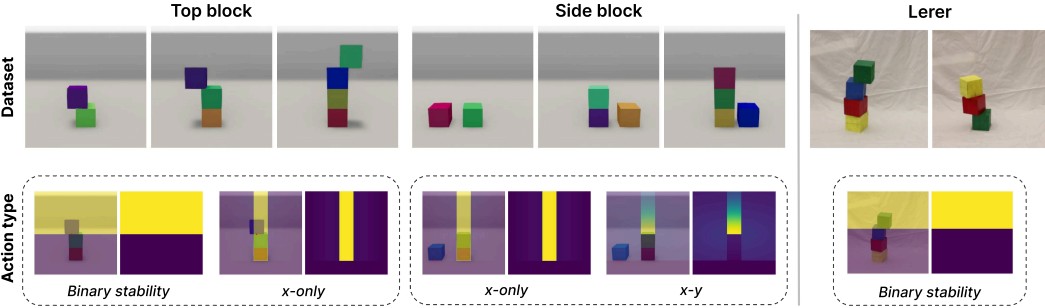

Figure 1: Overview of all combinations of datasets and action types. **Datasets:** We train models on two related datasets, one where the block on top of a tower is displaced, and one where a block is displaced on the ground next to a tower. **Action types:** For each dataset, we train models on two action types. *Binary stability* requires models to make a binary judgment on whether a given tower is stable. For the *x-only* task, models need to give a single value by which the displaced block should be moved to build a more stable or bigger tower. The *x-y* task requires the same action as *x-only*, but with an added dimension — here the block needs to be moved to the side and up. We train models on all four combinations of dataset and action types using SFT and GRPO to test whether models trained through interaction with an environment learn generalizable physical intuitions. We also evaluate models on an external dataset of real wooden block towers, taken from Lerer et al. (2016).

the model sequentially based on actions it has taken in an environment, whereas offline RL refers to updating the model based on a fixed set of state-action pairs collected using another policy (Levine et al., 2020). Online RL has been shown to outperform offline RL methods in some cases (Ostrovski et al., 2021), but this distinction has been under-explored in the context of VLMs. Chu et al. (2025) train VLMs on arithmetic reasoning and simple navigation tasks and find that online RL trained models generalize better than models trained with SFT. In contrast to Chu et al. (2025), our work focuses on established intuitive physics tasks in cognitive science: building and judging the stability of block towers (Lake et al., 2017).

## 3 METHODS

### 3.1 DATASETS

We construct two tower block datasets for our experiments, each consisting of stacks of 2-4 randomly colored cubes. 256×256 pixel RGB images are taken from a fixed camera angle in the ThreeDWorld environment (Gan et al., 2020). We keep the camera angle and block sizes fixed throughout, so that the models are able to learn the mapping between pixel space and ground truth distance. Both datasets feature towers that consist of perfectly stacked blocks except for one block. In the first dataset, **top block**, this block is on top of the tower but displaced to the left or the right (see Fig. 1 and Fig. 8 in the Appendix). In the second dataset, **side block**, the block is on the floor next to the tower, also either to the left or the right (see Fig. 1 and Fig. 9 in the Appendix). As an additional evaluation dataset, we also use an independent dataset of real images of tower blocks from Lerer et al. (2016; see Appendix A.1).

### 3.2 TASKS

Using these datasets, we construct four tasks. Using the top block and Lerer et al. (2016) datasets, the **binary stability** task requires the model to give a judgment on whether a given tower is stable or not. In contrast, the **x-only** task requires the model to return a single integer that moves the block along the $x$-axis (see Fig. 1). Here, the goal is to improve the stability of the tower by moving the block closer to the centre. In both tasks, the model must attend to the displacement of the top block from the centre point of the tower, both to judge its stability and to choose an appropriate counter-displacement to stabilize the tower. However, the latter case is framed interactively.

With the side block dataset, we also construct an **x-only** task, again requiring the model to give an integer to move the block to the most central position. In contrast, the **x-y** task requires the model to

give two integers to move the block to the most stable position in both the $x$- and $y$-dimensions (see Fig. 1). The x-only tasks are identical except that the range of correct integers is different due to the different block displacements. Moreover, models should be readily able to generalize from the x-y task to the x-only tasks, thanks to their being identical problems on the $x$-dimension. The prompts for each task are included in Appendix A.4.

### 3.3 FINE-TUNING METHODS

We fine-tune the 7B parameter 4-bit quantized version of the Qwen2.5-VL model (Yang et al., 2024; Wang et al., 2024) using the unsloth library (Han et al., 2023). We employ Parameter Efficient Fine-Tuning (PEFT; Han et al., 2024) — rather than updating all model weights we update small low-rank adapters inserted layer-wise in the model (QLoRA; Dettmers et al., 2024; Hu et al., 2021).

Two PEFT procedures are compared: reinforcement learning with Group-Relative Policy Optimization (GRPO), and Supervised Fine-Tuning (SFT). GRPO is our operationalization of the interactive condition, while SFT is our operationalization of the non-interactive condition. We outline each method in turn. Note that in Section 4.5.1 and Section A.8.4 in the Appendix, we also describe results from experiments with newer models and a different RL algorithm.

**Group-Relative Policy Optimization** In the reinforcement learning setting, the set of all model and adapter weights ($\theta$) is considered the policy, $\pi_\theta$. It takes the text prompt and image as input (observations of the state of the environment), and produces a token sequence as actions. For a batch of $M$ < prompt, image > pairs, $\{p_1, ..., p_M\}$, the model produces a set of $M \times N$ completions $\{c_{1,1}, ..., c_{1,N}, ..., c_{M,N}\}$. These completions are rewarded using a reward function, giving a set of rewards $\{r_{1,1}, ..., r_{1,N}, ..., r_{M,N}\}$. We use $N = 16$ in our experiments.

We compute the loss for some prompt $p$ as:

$$\mathcal{L}(\theta) = -\frac{1}{\sum_{i=1}^{N} |c_i|} \sum_{i=1}^{N} \sum_{t=1}^{|c_i|} [\min(\frac{\pi_\theta(c_{i,t}|q, c_{i,<t})}{\pi_{\theta_{old}}(c_{i,t}|q, c_{i,<t})} \cdot \hat{A}_{i,t}, \text{clip}(\frac{\pi_\theta(c_{i,t}|q, c_{i,<t})}{\pi_{\theta_{old}}(c_{i,t}|q, c_{i,<t})}, 1 \pm \eta) \cdot \hat{A}_{i,t})]$$

Where $|c_i|$ is the length of the completion, in tokens, and $\hat{A}_{i,t}$ is the normalized reward (advantage) for $|c_i|$:

$$\hat{A}_{i,t} = \frac{r_i - \text{mean}(\{r_1, ..., r_n\})}{\text{std}(\{r_1, ..., r_n\})}$$

Following common practice (Hu et al., 2025; Liu et al., 2025; Yu et al., 2025), we exclude the original KL-divergence term used in (Shao et al., 2024). We update the adapter weights with gradient ascent over $\mathcal{L}(\theta)$.

**Supervised Fine-Tuning** Using a labelled dataset, model weights are updated using batch gradient descent over the token-level cross-entropy loss:

$$\mathcal{L}(\theta) = -\sum_{t=1}^{T} \log p_\theta(y_t|y_{<t})$$

where $\theta$ is the set of model and adapter weights, $T$ is the ordered set of target completion tokens given a prompt, $y_t$ is the target token at step $t$, and $y_{<t}$ is the set of ordered target completion tokens prior to $t$. Only the adapter weight subset of $\theta$ are actually updated.

**PEFT Hyperparameters** We keep the hyperparameters across both fine-tuning methods as consistent as possible. All adapters are the same size and injected in all layers of the model. Specifically, we inject a matrix $W_a$ at each layer, which is the product of two low-rank matrices, $M_1 \in \mathbb{R}^{d \times r}, M_2 \in \mathbb{R}^{r \times k}$, where $d, k$ are the input and output dimensionalities respectively, and $r << d, k$. During the forward pass, the outputs of the full weight matrix for that layer are summed with the outputs of the adapter, subject to some scaling factor $\frac{r}{\alpha}$. We use $r = \alpha = 16$ everywhere.

We use stochastic gradient descent with the Adam optimizer. We train all models for 10,000 steps on single 80GB A100 GPUs.

### 3.4 REWARD FUNCTIONS

We use specific reward functions for each task both for training models with GRPO and for evaluating models in the results below. For the binary stability task, where models have to give a binary response judging the stability of a given block tower, we use three distinct reward values: $-1$ for non-parseable answers, $0$ for legal but incorrect answers, and $1$ for legal and correct answers.

For the x-only task where models reply with a single integer, we set the reward to $-5$ for non-parseable completions. For answers that are parsed correctly we use two different Gaussian functions based on the distance to the center. As answers get closer to the center, they are rewarded more. For answers that result in an unstable tower, we calculate a weaker function as $2 \cdot e^{(-d^2)} - 2$, where $d$ is the distance on $x$ from the optimal position. For answers that result in a stable tower, we compute the reward as $20 \cdot e^{(-d^2)}$ (see Fig. 1 and Fig. 8 in the Appendix for a visualization).

For the $x$-$y$ task where models must reply with two integers, we again set the reward to $-5$ for non-parseable answers. For answers that move the block below the floor, we set the reward to $-4$. For all other parseable answers, we again compute Gaussian reward functions depending on the euclidean distance between the final position of the moved block and the optimal position on top of the tower. For answers that are above ground but do not result in a stable bigger tower, we calculate the reward as $2 \cdot e^{(-d^2)} - 2$. For answers that are within the tower, we compute $2 \cdot e^{(-d^2)} - 4$. And for answers that result in a stable bigger tower, we compute the reward as $20 \cdot e^{(-d^2)}$ (see Fig. 1 and Fig. 9 in the Appendix for a visualization).

## 4 RESULTS

We evaluate performance on held-out instances from the post-training task (4.1), generalization to the other tasks (4.2), and generalization to the binary stability task after additional SFT (4.3).

### 4.1 POST-TRAINING PERFORMANCE IMPROVEMENT

Both SFT and GRPO improve performance of the pre-trained model on all post-training tasks (see the diagonal in Fig. 2). On the binary stability top block task, where models have to give a binary judgment on the stability of a block tower, the SFT and GRPO models achieve mean test accuracies of 0.936 and 0.92 after 10,000 steps (the ceiling here is 1, for all other tasks it is 20).

On the x-only top block task, models are asked to return a single integer to move the top block into a more stable position. Here, the SFT model achieves a mean test reward of 19.996 after 10,000 steps, and the GRPO model a mean reward of 19.995. For x-only on side block, models are asked to again return an integer to move a block, here one that is misplaced on the floor to the left or right of the tower, to the center of the image. The SFT model achieves a mean test reward of 16.774, whereas the GRPO model achieves a reward of 19.992. The x-y task on side block is similar to the x-only task, however models here also have to return a second integer to move the block to the left or right and up into the most stable position on top of the tower. Here, the SFT model gets a mean test reward of 17.527 and the GRPO model one of 19.868.

### 4.2 GENERALIZATION TO RELATED TASKS

We are not just interested in models that perform well on a single physical task, but rather models that robustly generalize from their experience to solve new tasks (Collins et al., 2022; Geirhos et al., 2018; Griffiths & Tenenbaum, 2009). Therefore, we test whether models post-trained on single tasks with either method can generalize to new, related tasks. To test this, we evaluate all fine-tuned models on all tasks.

We find that no model reliably generalizes to other tasks, regardless of the fine-tuning method (see Fig. 2). The only models that show some generalization are the SFT and GRPO models trained on the x-y side block task. These models also perform well on the x-only side block task, with

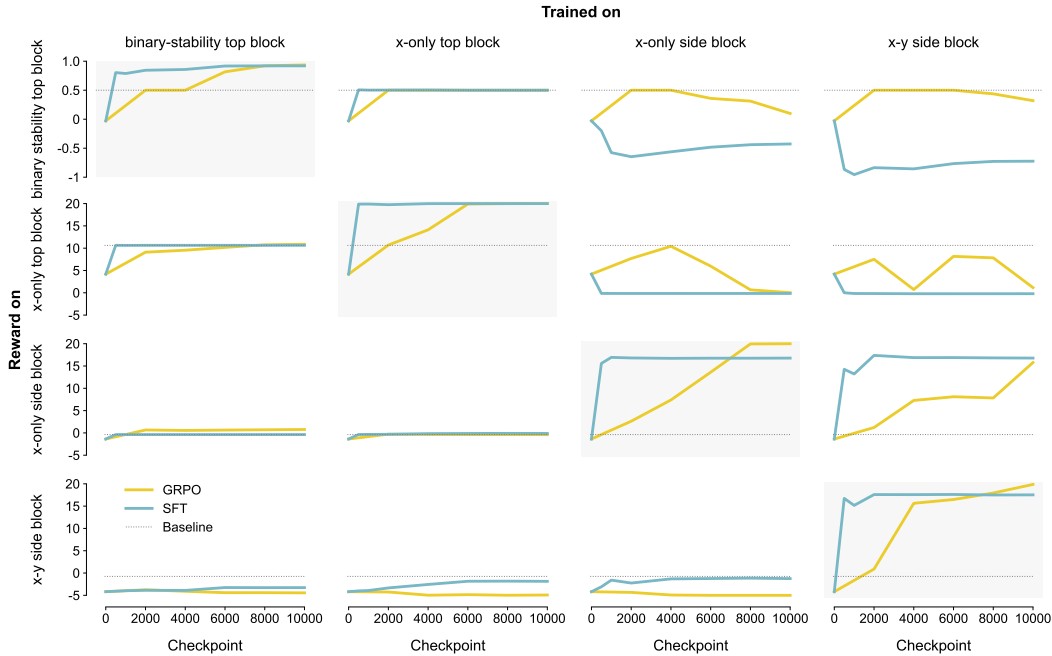

Figure 2: Performance by test task and training task. Rows show models evaluated on a given task. Columns show models trained on a given task. The blue and orange lines show the performance of the models trained with SFT and GRPO, respectively. The grey dotted line shows the baseline for the evaluation task. Plots on the diagonal show within-task performance, meaning models are evaluated on the same task they are trained on. All other subplots represent some degree of generalization.

the GRPO model achieving a mean reward of $15.781$ and the SFT $16.778$. This is to be expected because x-only is a subset of the x-y task which only requires the output of the x variable that the model has learned. In contrast, the GRPO models trained on binary stability top block and x-only top block get mean rewards of $0.767$ and $-0.396$, and the SFT models $-0.372$ and $-0.123$ on x-only side block (which is around the baseline for an agent that always leaves the block in its initial position).

Another condition where we expected strong carry-over was between binary stability and x-only on top block, as solving both tasks requires the same task variable, the x-offset of the top block. In the x-only condition, the model is explicitly forced to learn this variable as it is also the amount the top block has to be moved by in order to put it in the most stable position. It is also the single variable needed to solve whether a given block tower is stable or not. Despite this, we find no generalization between the two conditions, showcasing how constrained generalization from either post-training method is. When evaluated on binary stability top block, the GRPO models trained on x-only top block, x-only side block, and x-y side block get accuracies of $0.501$, $0.0989$, and $0.32$ respectively. The SFT models trained on the same conditions perform at $0.5$, $-0.427$, and $-0.724$ (since this is a binary task the random baseline is $0.5$, but illegal answers can pull the reward down).

### 4.3 ADDITIONAL SUPERVISED FINE-TUNING

It is possible that the models have learned some task-general features, but that they fail to perform well on new tasks due to some task-specific properties. To test this, we take checkpoints of the GRPO and SFT x-only top block models, and fine-tune them on the binary stability top block task with some additional steps of SFT. If the models have learned some task-general features, they should learn the binary stability task more quickly than the base model — this means that they should require fewer additional fine-tuning steps to reach good performance.

We see that the models very quickly reach high accuracies in the binary stability task after just a few steps of supervised fine-tuning (see Fig. 3). In comparison, the base model fine-tuned with the same number of steps performs less well — while it takes 25 steps for the base model to reach the random

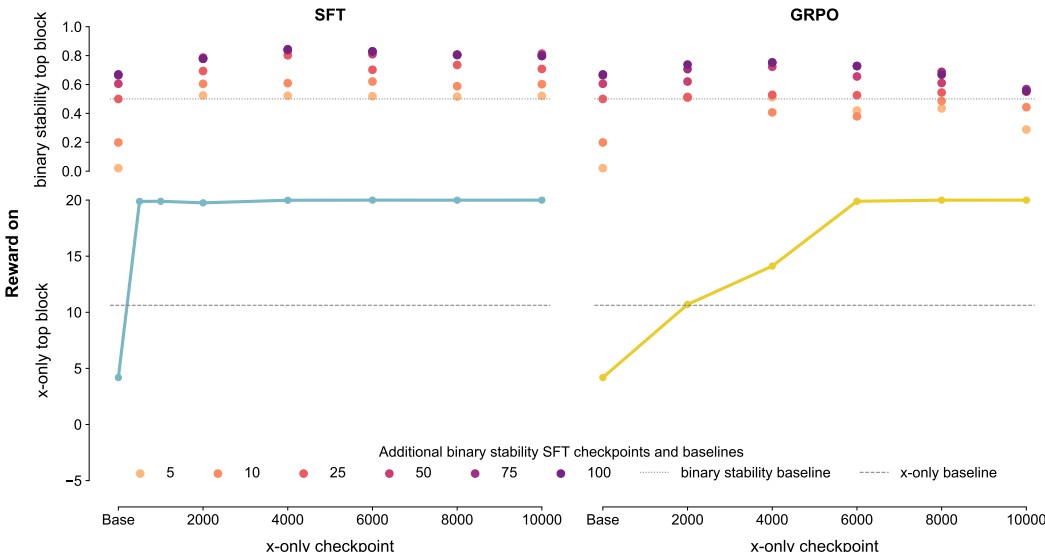

Figure 3: Generalization after additional post-training. The blue line on the bottom left shows the test performance of the SFT-trained model and the orange line on the bottom right shows that of the GRPO-trained model. Both models were trained on the x-only top block task. For each checkpoint, we take the model trained on this task up to that checkpoint and train it with SFT on the binary stability top block task for up to 100 additional steps. The stacked dots on the top show different checkpoints of the additional binary stability trained models, based on the x-only model at that checkpoint, and the reward they achieve on the binary stability task.

baseline for this binary task, all SFT x-only top block checkpoints are already over the baseline after 5 additional SFT steps on binary stability. This is likely in part because the base model has not yet learned to format its answers correctly, whereas all post-trained models have experience with the correct answer format. However, after 100 steps of SFT on binary stability, the base model only returns legal answers but still achieves a lower accuracy than the post-trained models with the same number of additional SFT steps, meaning formatting can not explain the whole performance gap.

We see that SFT post-trained models in general achieve slightly higher accuracies than their GRPO counterparts after 100 additional steps of SFT on the binary stability task: the models trained with SFT on x-only top block up to 2000, 4000, 6000, 8000, and 10000 steps achieve accuracies of 0.778, 0.844, 0.829, 0.806, and 0.800. In contrast, the same checkpoints for the GRPO trained models get accuracies of 0.749, 0.695, 0.745, 0.659, and 0.588. We also see that GRPO models from earlier x-only checkpoints are able to achieve higher accuracies on the binary stability task than later checkpoints.

### 4.4 DECODABILITY ANALYSIS

To further explore whether the models have learned some task-general features, we analyzed activations during the forward pass to see if the models represent the information necessary to generalize to our set of intuitive physics tasks (see section A.6 in the Appendix for more details). If the activations encode this information, it suggests that the models have the competence to solve the intuitive physics tasks, but fail to convert that into good performance.

We find that the binary stability of a tower is already trivially decodable in the base model, and it does not change in any substantial way through either fine-tuning method (see Fig. 11 in the Appendix). This is likely because there exists an obvious pixel level shortcut where tower stability can be determined from a small set of pixels along the horizontal center line in the image. However, while this information is decodable from everywhere throughout the model, the base model performs the binary stability task at much lower accuracies than achieved by the linear probes.

The same is true for the offset of the top block. Again, already in the base model, the x-offset is highly decodable. The decodability of the x-offset is higher in the language layers for models trained

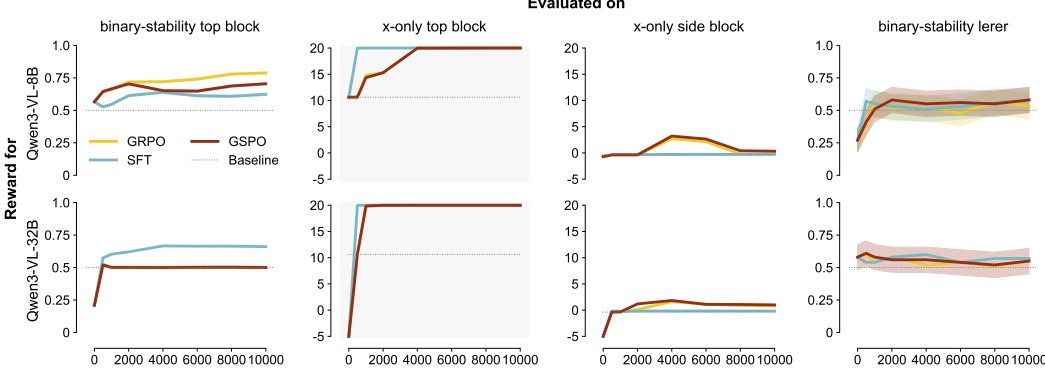

Figure 4: Qwen3-VL-8B and Qwen3-VL-32B, trained on *x-only top block*. Crucially, we find that these models show some generalization to the related *binary-stability top block* task. These two tasks share the exact same data and they also share the same properties of interest, with the offset of the top block being the necessary property to solve both tasks. However, we still find that they do not generalize to the other tasks, such as *x-only side block*. Furthermore, models also do not generalize to judging the stability of real block towers from Lerer et al. (2016). Error bars show 95% confidence intervals.

with GRPO compared to SFT and the base model, although these differences are small. This analysis suggests that while the models have the competence to solve either task, this does not translate to good performance, hinting at shortcut learning. This invites the question of what information the models use when solving these tasks. To better understand this, we look at what they attend to in an image. We compare the attention maps of post-trained models to those of the base model to see if they learned to focus on specific blocks or positions in the image. However, these attention maps are noisy and do not reliably show any differences in model attention as a result of either post-training method. Some examples are shown in Appendix A.7.

### 4.5 FOLLOW-UP EXPERIMENTS

We perform a number of follow-up experiments to ensure our results are not simply artifacts of the training duration, the generation length, or the adapter rank. We report these results in Section A.8 of the Appendix. We also evaluate newer and bigger model variants, as well as models trained with another RL algorithm in Section 4.5.1 below. Additionally, we also test if models can generalize to real images and whether training on multiple tasks leads to better generalization in Section 4.5.2 below.

#### 4.5.1 OTHER MODELS AND RL IMPLEMENTATIONS

To ensure our results transfer from Qwen2.5-VL-7B to other models, we post-train Qwen3-VL-8B and Qwen3-VL-32B on the *x-only top block* task with SFT and GRPO. To ensure that our results transfer to other RL algorithms, we also post-train these newer models with Group Sequence Policy Optimization (GSPO). GSPO replaces the token-level optimization in GRPO with sequence-level optimization (see Zheng et al. (2025) for more information). Again, we find that these models perform well on their post-training task, *x-only top block*. However, we also find that they show some traces of generalization to the related *binary-stability* task (see Fig 4). These two tasks share the same data and physical properties of interest: the offset of the top block. However, we still find no generalization to the other tasks, such as for example *x-only side block* (see also Figs. 16 and 17 in the Appendix). This task is the same as the models' post-training task, only with larger block displacements. If the models learned the mapping between block distance to center and the integer action space, they should in principle also be able to solve this task — but we do not find this. Additionally, as outlined in the section below, we tested these models on judging the stability of real block towers from Lerer et al. (2016). We find that no model performs well on this task, regardless of model generation, model size, and if they are trained with interaction or not (see Fig. 4 and Fig. 18 in the Appendix).

### 4.5.2 GENERALIZATION TO REAL IMAGES AND FROM MORE VARIED FINE-TUNING

To test whether models learn physical intuition that can generalize to real images, we test them on 100 images from Lerer et al. (2016). These images show real images of wooden block towers, which are either stable or unstable. We again find that no model performs well on this task — even the model trained on our similar *binary-stability top block* stimuli (see Figs. 7 and 19 in the Appendix).

As outlined in previous work (Schulze Buschoff et al., 2025a), it is possible that models trained on a single task fail to generalize because they are not exposed to enough variance in their post-training. To check if generalization can be improved by incorporating multiple tasks, we train a GRPO model first on *x-only side block* and then on *binary-stability top block*. This ensures that the model has been exposed to two different tasks and data sets. Since it has been trained on the x-only task and also on the top block data set (albeit not at the same time) we would expect this model to generalize well to the *x-only top block* task. We find that this model is still able to perform both tasks it was trained on (see Fig. 15 in the Appendix). While it has some trouble keeping the correct formatting for the first task it was trained on, filtering only legal answers reveals that it still retains the capacity to solve it.

We also train an SFT model in the same blocked manner, first training it on *x-only side block* and then on *binary-stability top block*. This model quickly degrades in performance on the task it was trained on first (see Fig. 15 in the Appendix). Additionally, we also train a SFT variant with interleaved data, training it on *x-only side block* and *binary-stability top block* at the same time. In contrast to the blocked SFT model, this joint model performs reasonably well on both of its post-training tasks.

## 5 DISCUSSION

Recent evidence has suggested that vision language models (VLMs) do not have robust human-like intuitions about the physical world. For instance, they struggle to reason about the stability of block towers or about cause and effect (Schulze Buschoff et al., 2025a), even when they are fine-tuned on related tasks (Schulze Buschoff et al., 2025b). Humans, on the other hand, have robust intuitions about the physical world, which they learn in part from interacting with their environment.

To capture this aspect of human learning, we trained VLMs on intuitive physics tasks that require building block towers through interaction with an environment. We trained these models using the online reinforcement learning algorithm Group-Relative Policy Optimization (GRPO), and compared it to Supervised Fine-Tuning (SFT), an analogue of offline reinforcement learning (Levine et al., 2020). We then tested these trained models on held-out tower building tasks and on judging the stability of block towers.

Given the relevance of interaction for learning intuitive physics, we defined three hypotheses: (1) that GRPO-trained models would outperform SFT-trained models on held-out instances of the task they were trained on; (2) That GRPO-trained models would generalize better than SFT-trained models to new tasks, such as judging tower stability; (3) That GRPO-trained models would learn more quickly to accurately judge tower stability than SFT-trained models.

Our experiments found no evidence in favor of (1). Across all four tasks, both GRPO and SFT post-training led to models performing close to ceiling on held-out test instances. SFT post-training led to ceiling performance within 500-1000 steps, while GRPO post-training took between 4000 and 8000 steps to achieve good performance. This supports recent results showing that task-specific post-training can make VLMs perform well within the visual intuitive physics problems they are trained on (Balazadeh et al., 2024; Schulze Buschoff et al., 2025b).

In contrast, for (2), we found that neither post-training method conferred a clear advantage for generalizing to new tasks in a zero-shot setting. To start, the Qwen2.5-VL-7B models did not generalize between any two tasks, even though our decodability analysis showed that the properties of interest, such as both binary stability and x-offset, are highly decodable from activations at all intermediate layers in the base, SFT-trained and GRPO-trained Qwen2.5-VL-7B model. Yet, neither post-training method was able to make the model use this information on out-of-distribution tasks. In contrast, the newer Qwen3-VL variants showed traces of generalization between the binary stability and x-only tasks on the top block dataset. Both tasks require the model to attend to the same quantity, namely, the degree to which the top block is offset from the center of the tower. However, we still find

that they do not generalize to the other tasks. Furthermore, they also do not generalize to stability judgments of real block towers. To summarize, while we find some traces of generalization between the x-only top block and binary-stability top block tasks in these newer models, it is very limited and the models do not transfer to other related tasks. We hypothesized that interaction would be helpful for learning generalizable physical intuitions. However, we again do not find clear evidence that training these models with interaction gives them generalizable physical intuitions, nor that it is better than SFT when it comes to generalization to related tasks.

For (3), we find that models post-trained with either method generalize to the binary stability task more quickly than the base model after the same number of SFT steps. Both GRPO and SFT models trained on x-only, given some additional steps of SFT on binary stability, perform above chance on held-out binary stability tests. This suggests that training on tower building first can help models learn to judge tower stability more quickly — the models seem to, at least in part, learn some generalizable features. However, the GRPO-trained model is slightly worse for each checkpoint compared to its SFT counterpart. This is likely because the GRPO model learns to produce extra tokens alongside its answers, which are immediately penalized during SFT post-training on the binary stability task.

In summary, interaction, in the sense of one-step reinforcement learning with GRPO, does not appear to confer a general advantage for solving a family of related intuitive physics problems. Indeed, GRPO appears to perform similarly within-distribution to SFT, which is also unable to facilitate reliable generalization. These results, along with our decodability analysis, indicate that models trained with either post-training method learn non-general shortcuts rather than robust physical intuitions.

There are several avenues of research that would strengthen the conclusions of this study. We investigate only three models of sizes 7B, 8B, and 32B, using relatively small quantities of data. Future work will examine whether our conclusions apply to larger models trained on larger volumes of data. We also only investigated 1-step interactions with the environment. It remains possible that advantages of interaction only surface when models are able to interact with their environment over long state-action sequences. Future work will investigate practical methods for testing this with modern vision-language models. Finally, we found evidence that some newer models (Qwen3-VL) trained with specific algorithms and hyperparameters show limited generalization between tasks. This result merits further investigation to determine the robustness and extent of this generalization on intuitive physics tasks.

## 6 CONCLUSION

We hypothesized that through interaction with an environment, vision language models would be able to learn generalizable physical intuitions. However, we find little evidence of this — neither models trained with GRPO nor SFT were able to generalize from their training task to a new task, even when that task required attention to the same physical quantity on the same visual stimuli. This suggests that these models are not learning true physical intuitions, but rather task-specific shortcuts.

Together, our results suggest that prominent post-training methods are constrained in the ways that they can improve models when it comes to intuitive physics. It remains unclear whether post-training models on specific cognitive tasks is sufficient for developing models that reason about the world in a human-like manner. Developing machine learning models with these abilities may require different pre- and post-training paradigms that go beyond parameter-efficient adaptation.

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

# A APPENDIX

## A.1 DATA EXAMPLES

### A.1.1 TOP BLOCK DATASET

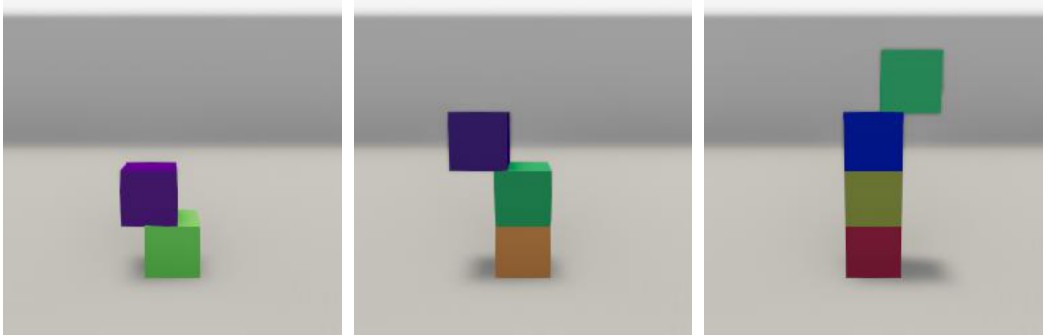

Figure 5: Example images for the *top block* dataset. Images feature towers with 2 to 4 blocks with the top block displaced to the left or the right on top of the tower.

### A.1.2 SIDE BLOCK DATASET

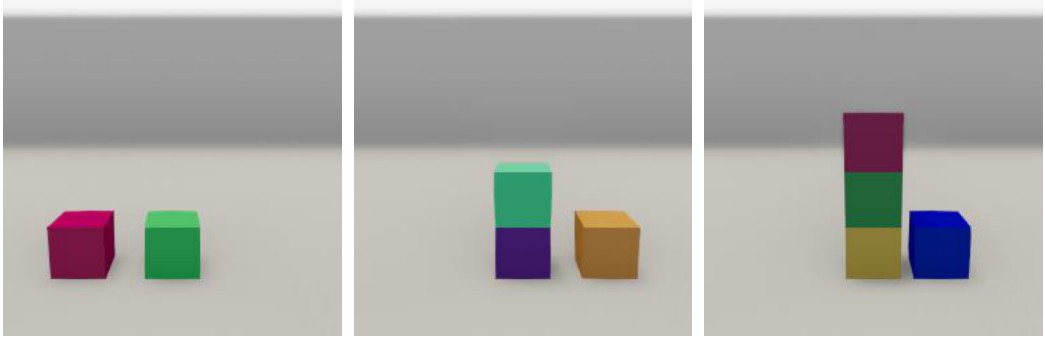

Figure 6: Example images for the *side block* dataset. Images feature towers with 1 to 3 blocks with a misplaced block to the left or the right side of the tower.

### A.1.3 LERER DATASET

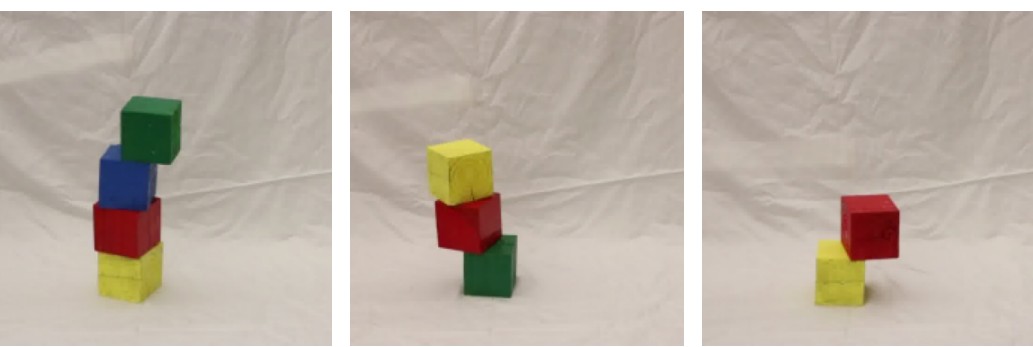

Figure 7: Example images for the *Lerer* evaluation dataset. Images are real pictures of block towers with 2 to 4 blocks with the upper blocks displaced to the left or the right.

## A.2 MAIN RESULT TABLES

The following table shows the results shown in Fig. 2 for all models after 10.000 steps of GRPO:

| Evaluated on | Trained for 10.000 steps with GRPO on | | | |
|---|---|---|---|---|
| | x-only binary stability | x-only top block | x-only side block | x-y side block |
| x-only binary stability | 0.936 | 0.501 | 0.100 | 0.320 |
| x-only top block | 10.881 | 19.995 | 0.026 | 1.161 |
| x-only side block | 0.767 | -0.373 | 19.992 | 15.782 |
| x-y side block | -4.448 | -4.926 | -5.000 | 19.870 |

The following table shows the results shown in Fig. 2 for all models after 10.000 steps of SFT:

| Evaluated on | Trained for 10.000 steps with SFT on | | | |
|---|---|---|---|---|
| | x-only binary stability | x-only top block | x-only side block | x-y side block |
| x-only binary stability | 0.920 | 0.500 | -0.427 | -0.724 |
| x-only top block | 10.650 | 19.288 | -0.155 | -0.206 |
| x-only side block | -0.373 | -0.123 | 18.572 | 16.778 |
| x-y side block | -3.268 | -1.862 | -1.251 | 15.839 |

## A.3 REWARD FUNCTION VISUALISATION

Below, we visualize the reward functions for the *x-only* and *x-y* tasks. For *x-only*, the reward for answers that result in an unstable tower is calculated as $2 \cdot e^{-d^2} - 2$, where $d$ is the distance on the $x$-dimension. For answers that result in a stable tower, we compute the reward as $20 \cdot e^{-d^2}$ (see Fig. 8 below).

For *x-y*, we compute the euclidean distance between the final position of the moved block and the optimal position on top of the tower. For answers that are above ground but do not result in a stable bigger tower, we calculate the reward as $2 \cdot e^{-d^2} - 2$. For answers that are within the tower, we compute $2 \cdot e^{-d^2} - 4$. And for answers that result in a stable bigger tower, we compute the reward as $20 \cdot e^{-d^2}$ (see Fig. 9 below).

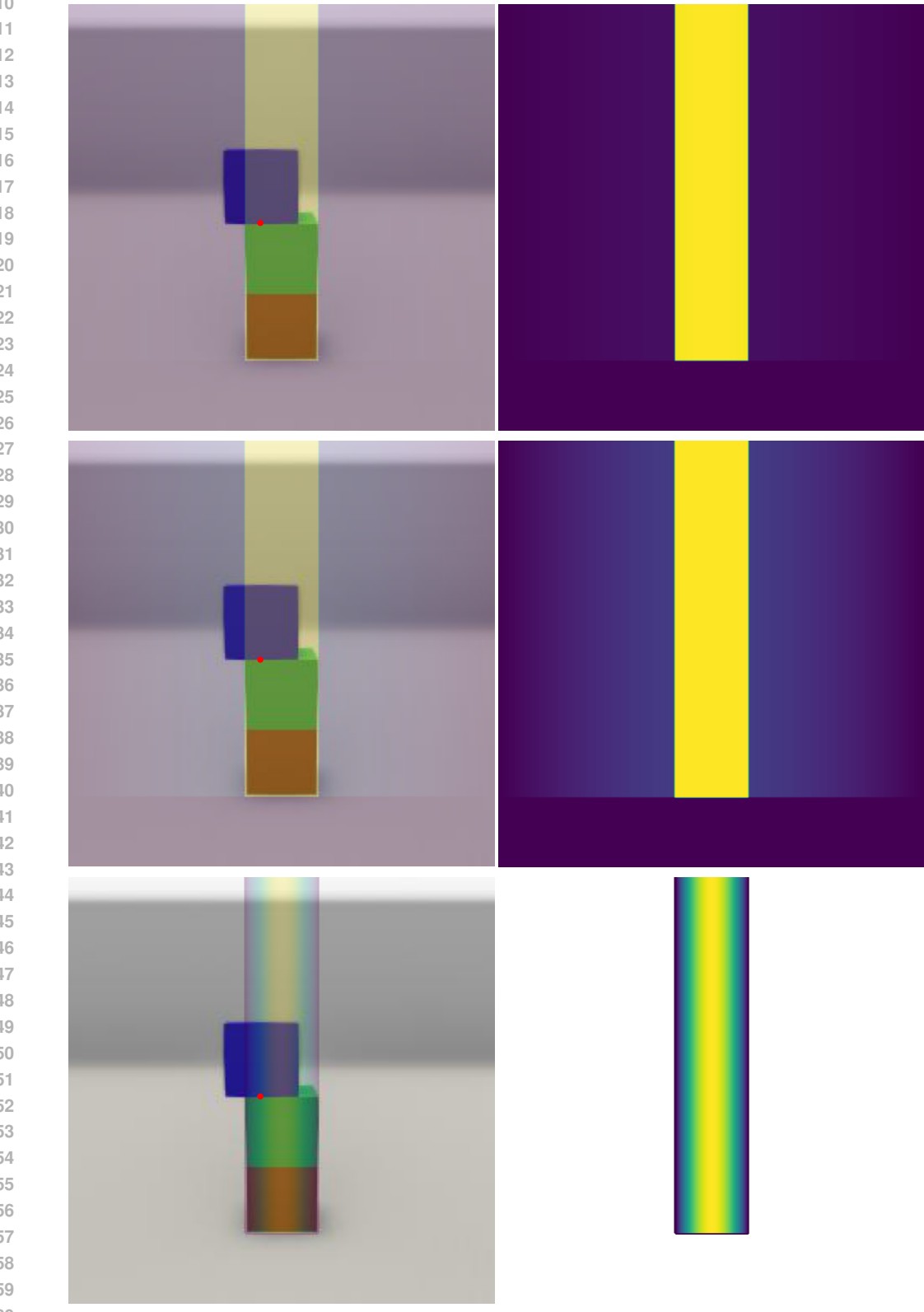

Figure 8: Reward function for the x-only task on the top block dataset. The red dot sits at the lower center of the block from which the reward is calculated. The first row shows the non normalized reward values. The second row shows the symmetric-log transformed reward values. The third row shows the log normalized reward values.

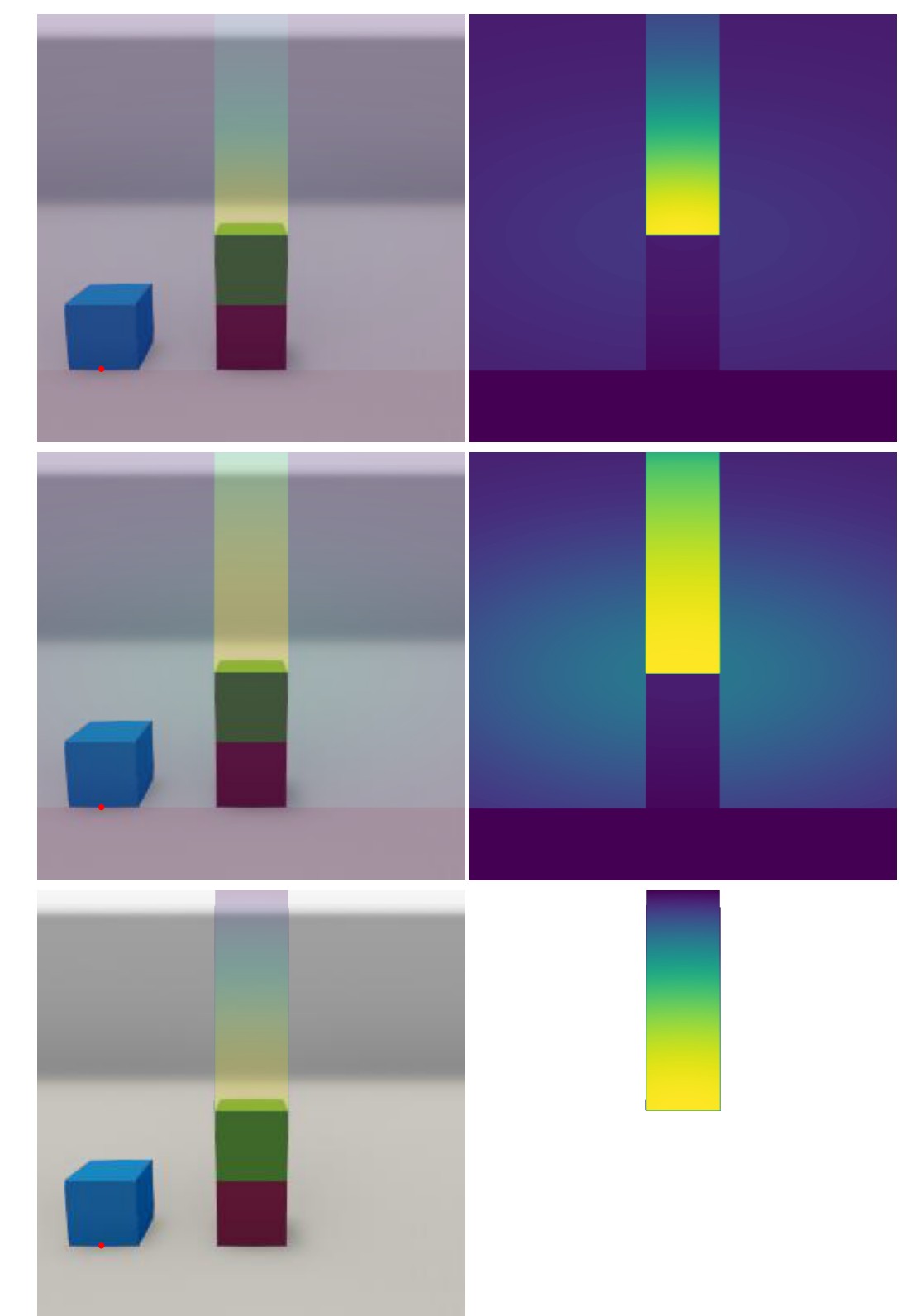

Figure 9: Reward function for the x-y task on the side block dataset. The red dot sits at the lower center of the block from which the reward is calculated. The first row shows the non normalized reward values. The second row shows the symlog transformed reward values. The third row shows the log normalized reward values.

A.4 Task Prompts

We use the following prompt variations for the four tasks depending on action type and dataset (see Fig. 1 for an overview):

For *binary judgment* on the *top block* dataset:

> In the image you see a block tower with a single misplaced block on top. Your task is to determine if this is a stable tower. Respond with Yes if the tower is stable or No if it is not stable. Return your final answer between <answer> </answer>.

For *x-only* on the *top block* dataset:

> In the image you see a block tower with a single misplaced block on top. Your task is to build a stable tower by moving the top block to the most stable position. You can move the top block to the left or right, by responding with an integer between -600 and 600. Return your final answer between <answer> </answer>.

For *x-only* on the *side block* dataset:

> In the image you see a block tower in the centre with a single misplaced block to the side. Your task is to build a stable tower by moving the misplaced block to the most stable position on the top of the tower. You can move the top block to the left or right, by responding with an integer between -600 and 600. Return your final answer between <answer> </answer>.

For *x-y* on the *side block* dataset:

> In the image you see a block tower in the centre with a single misplaced block to the side. Your task is to build a stable tower by moving the misplaced block to the most stable position on the top of the tower. You can move the misplaced block to the left or right and up, by responding with two integers. The first integer should be between -600 and +600 and moves the block left or right. The second integer should be between 0 and +1000 and moves the block up. Return your final answer between <answer> </answer>.

For *binary-stability* on the *Lerer* dataset:

> In the image you see a block tower. Your task is to determine if this is a stable tower. Respond with Yes if the tower is stable or No if it is not stable. Return your final answer between <answer> </answer>.

For *x-only* on the *top block* dataset with reasoning (long generation):

> In the image you see a block tower with a single misplaced block on top. Your task is to build a stable tower by moving the top block to the most stable position. You can move the top block to the left or right, by responding with an integer between -600 and 600. Provide your reasoning between <think> and < /think>. You can think about the problem for as long as you'd like. While thinking, you should robustly verify your solution. Return your final answer between <answer> </answer>.

## A.5 TRAINING LOGS

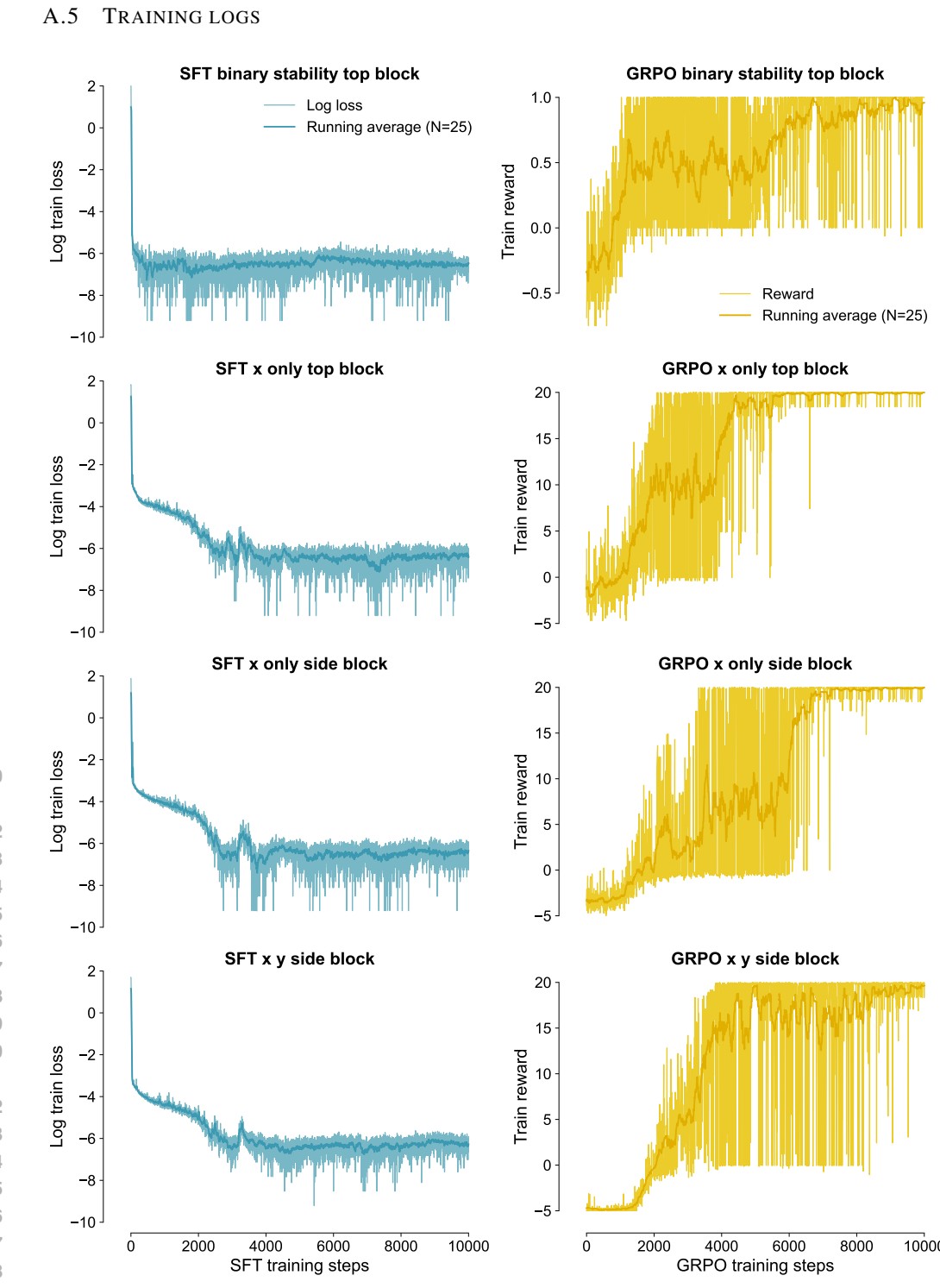

Figure 10: Training logs for the SFT (left) and GRPO (right) Qwen2.5-VL-7B models trained on all datasets. For the SFT models, we show the log loss and a running average with a window of 25. For the GRPO models, we show the mean reward and a running average with a window of 25.

## A.6 DECODING ANALYSIS

For the top block dataset, we train linear probes on the representation of the model at each layer to predict the binary stability of a tower and the x-offset of the top block from those representations. Since the image tokens appear before the text tokens, the linear probes only have access to the visual information. We run this process with 10-fold cross validation using 600 images in total. For the binary stability analysis, we train L2 regularized logistic regression models on the representations. For the x-offset analysis, we train linear regression models with spherical Gaussian priors.

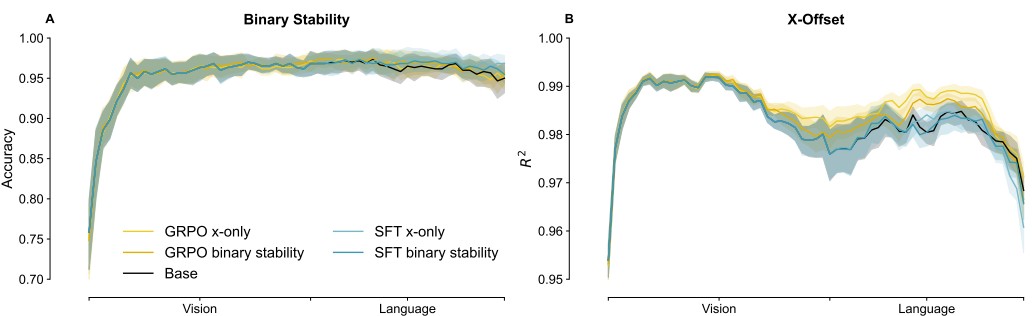

Figure 11: Physical property decodability analysis. (A) shows the decodability of the ground truth binary stability for the base model, as well as models post-trained with GRPO and SFT on x-only top block and binary stability top block. (B) shows the decodability of the x-offset of the uppermost block in the top block dataset for the same set of models.

## A.7 ATTENTION MAPS

To better understand how finetuned models learn to solve our tasks, we compared the attention maps of the fine-tuned models and the base model. More specifically, our goal was to provide a qualitative comparison of how much the last token in the question prompt attends to the different image tokens, throughout the layers of the language model. In Fig. 12, we show the attention maps averaged across heads for each layer for both the base model and a GRPO model post-trained on x-only top block. As seen in the example below, there is no clear change as a result of the post-training method that would give us a better understanding of the strategy used by the post-trained models.

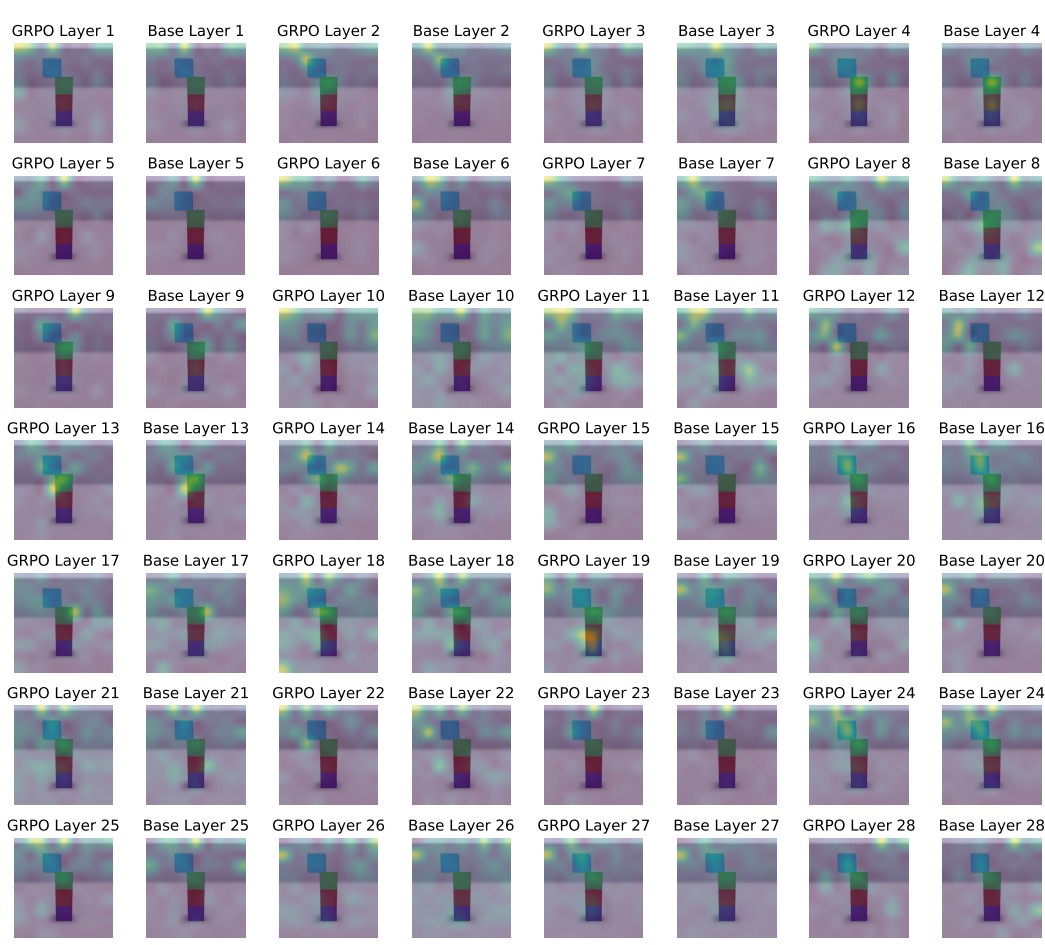

Figure 12: Attention maps for the base model and the model post-trained with GRPO on x-only top block. The model is asked if a given tower is stable or not. Attention maps over the different heads are averaged in each layer.

## A.8    ADDITIONAL CHECKS

### A.8.1    LONGER TRAINING HORIZON

We find that training models with GRPO for longer only leads to overfitting to the specific training task (see Fig. 13). As training exceeds 10.000 steps, models tend to overfit too strongly to the specific reward function of the training task to generalize to other tasks — while we still saw some generalization for the x-y side block trained model to the x-only side block task, this disappears as the models are trained for longer. The results reported above all use a restricted generation length due to resource constraints.

To test whether generalizable physical intuitions could emerge in GRPO models over time, we trained them for up to 48.000 steps. We find that as we exceed 10.000 steps, the model tends to overfit too strongly to the specific reward function of the training task to generalize to other tasks — while we still saw some generalization for the *x-y side block* trained model to the *x-only side block* task, this disappears as the models are trained for longer.

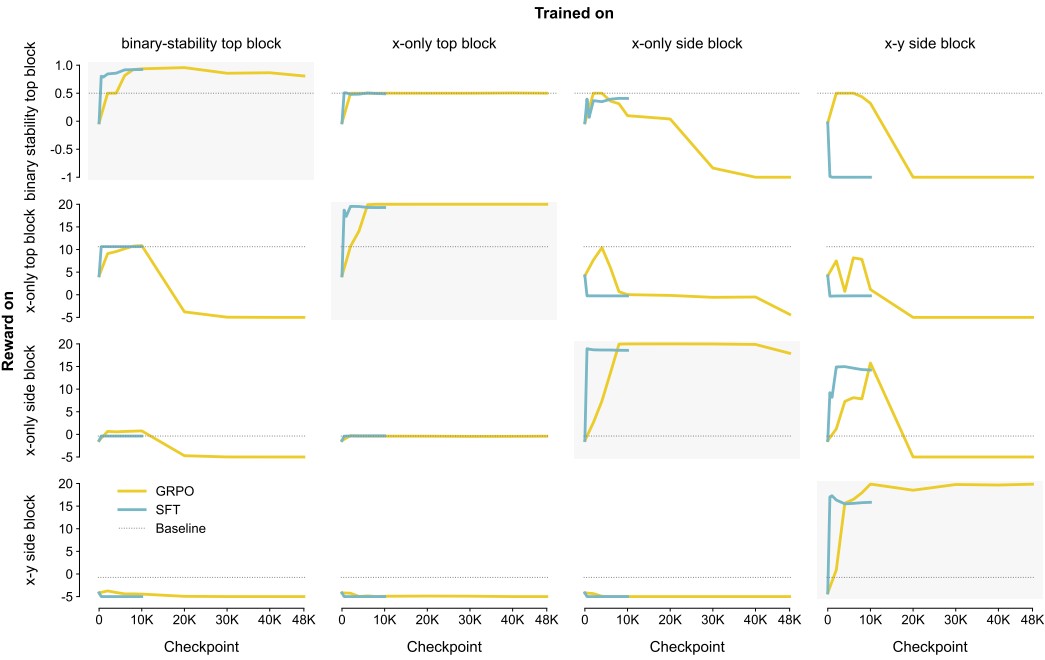

Figure 13: Performance for a longer horizon. We show the same plot as 2 but with results for up to 48K training steps for the GRPO models. Performance is shown for each combination of test task and training task. Rows show models evaluated on a given task. Columns show models trained on a given task. The blue and orange lines show the performance of the models trained with SFT and GRPO, respectively. The grey dotted line shows the baseline for the evaluation task. Plots on the diagonal show within-task performance, meaning models are evaluated on the same task they are trained on. All other subplots represent some degree of generalization.

### A.8.2 TRAINING ABLATIONS

To test if allowing the model to reason about the task for longer improves generalization, we train a model on the *x-only top block* task with a longer generation length (see A.4 for the reasoning prompt). However, as shown in Fig. 14, we find that this model also does not generalize to the other tasks.

The results we report use a default rank of 16 for all models. To make sure that this does not cause the models to overfit to our task specifically, we train models on the *x-only top block* task using ranks of 1 and 8 instead. We find that these models show the same failure to generalize. While models of all ranks learn to perform well on their training task, they do not generalize to any other task (see Fig. 14 in the Appendix).

Additionally, to ensure that the models do not suffer from overfitting the vision encoder, we train a model with the standard rank of 16 but without fine-tuning the vision encoder. We find that this model shows a similar performance over all tasks as the model with vision fine-tuning, again not generalizing to other related tasks from the training task (see Fig. 14 in the Appendix).

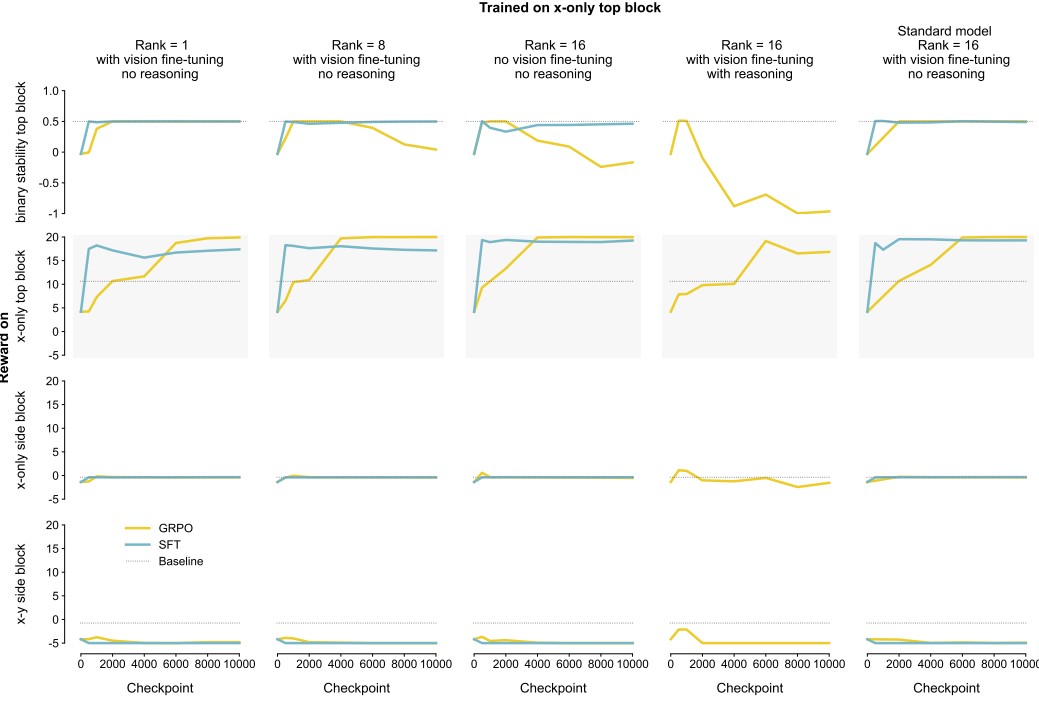

Figure 14: Ablation checks. All models are trained on x-only top block but they have lower ranks (1 and 8 compared to 16) or are trained without fine-tuning of the vision encoder or they are trained with reasoning (a larger generation length). All ablations learn to perform well on the task they are trained on (shown in the second row). However, all models fail to generalize to other related tasks — just as the standard model we used throughout our experiments (last column).

### A.8.3 BLOCKED AND INTERLEAVED JOINT TRAINING

To test whether models could generalize if they are exposed to multiple tasks at the same time, we trained models on two tasks: *x-only side block* and *binary-stability top block*. Since this model has seen the *x-only* task and also the *top block* data set (albeit not at the same time), it should be able to generalize to the *x-only top block* data set. We show results for this in the figure below.

We find that the GRPO model that has been trained on both tasks can still perform both tasks. The model has some trouble keeping the correct formatting for the first task block it was trained on, but filtering only legal answers reveals that it still retains the capacity to solve it. The SFT model on the other hand quickly degrades in performance on the task it was trained on first. This indicates

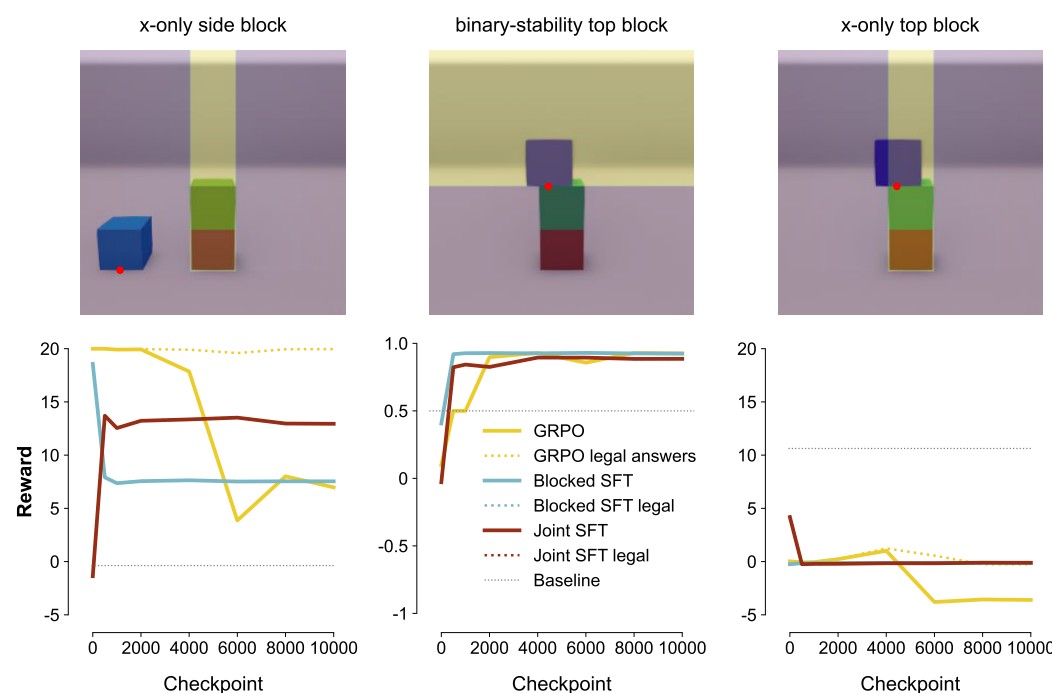

Figure 15: Blocked joint training. The model was first trained for 10.000 steps on *x-only side block*. It is then trained for 10.000 steps on *binary-stability top block* and performance is shown below over these 10.000 steps. The model forgets the proper formatting of responses for the initial *x-only side block* task (see continuous line on the left), but legal answers still perfectly solve the task (see dotted line on the left). The model can perform both tasks it was trained on, however it does not generalize to the *x-only top block* task, even though it has seen both the *x-only* task and also the *top block* data set (albeit not at the same time).

some benefit of GRPO when training models on multiple tasks successively. However, the joint SFT model that is trained on both tasks at the same time, in contrast to in a blocked manner, can overcome this shortcoming, performing reasonably well on both of its post-training tasks.

### A.8.4 NEWER AND BIGGER MODELS AND ANOTHER RL IMPLEMENTATION

To test whether our results generalize to other models, we train Qwen3-VL-8B and Qwen3-VL-32B with GRPO and SFT on the *x-only top block* task. We show results for this in the figures below. To test whether our results generalize to other models and RL implementations, we also train these newer models with GSPO on the *x-only top block* task. We show results for this in the figures below.

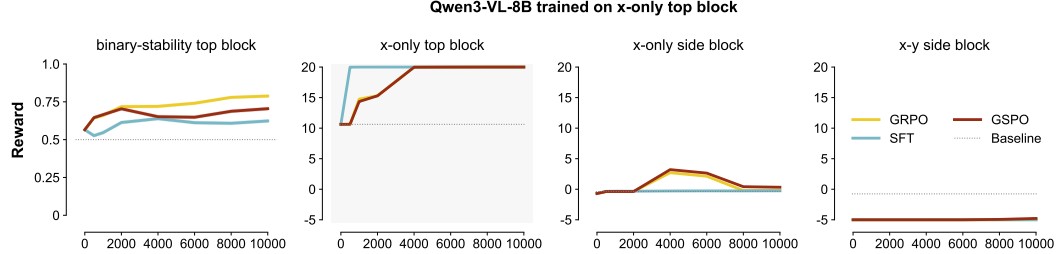

Figure 16: Qwen3-VL-8B model trained with GRPO, GSPO, and SFT on the *x-only top block* task. The model also does not generalize from its' fine-tuning task to other related tasks. Noticeably, the model is above chance for the *binary-stability top block* from the get-go and improves slightly over the course of training on the related *x-only top block* task under all post-training regimes.

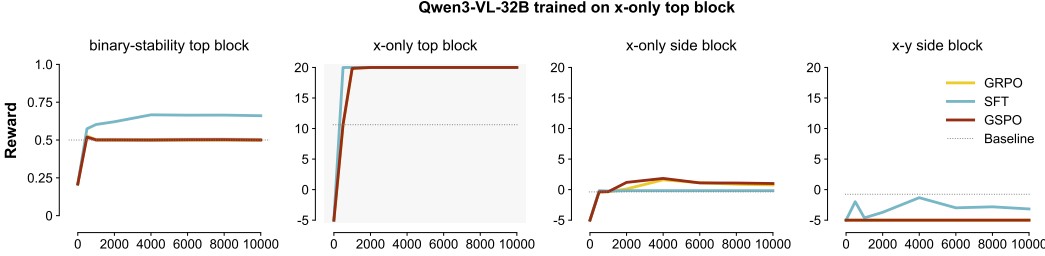

Figure 17: Qwen3-VL-32B model trained with GRPO, GSPO, and SFT. The model is trained on the *x-only top block* task. Noticeably, the SFT post-trained model improves slightly over the course of training on the related *binary-stability top block*. In contrast, the GRPO/GSPO post-trained models do not generalize from the *x-only top block* task to the *binary-stability top block* task.

### A.8.5  GENERALIZATION TO REAL IMAGES

To test whether models could generalize to the same task but presented in natural images, we test all models on 100 images from Lerer et al. (2016). The model trained on *binary-stability top block* is trained on the same task — predicting binary stability for block towers — but still does not generalize to these images.

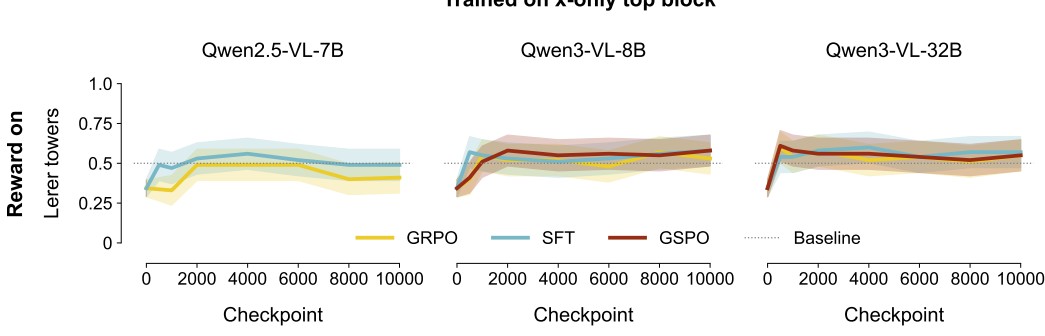

Figure 18: All models trained with GRPO, GSPO, and SFT on the *x-only top block* task, evaluated on the real block towers from Lerer et al. (2016). While we still found some generalization from post-training on our *x-only top block* task to our *binary-stability top block* task, the models do not generalize to this external task. Error bars show 95% confidence intervals.

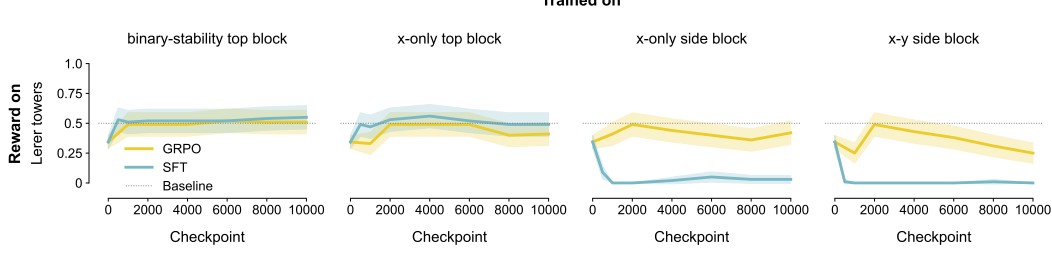

Figure 19: Qwen2.5-VL-7B models trained on all four tasks, evaluated on the real block towers from Lerer et al. (2016). We again find that no model performs well on this task — even the model trained on our similar *binary-stability top block* task. Error bars show 95% confidence intervals.

