# OpenReview forum: "Can vision language models learn intuitive physics from interaction?"
_ICLR.cc/2026/Conference — Submitted to ICLR 2026_

### Official Review · Reviewer_aHQU · 2025-10-15

**Soundness:** 2
**Presentation:** 2
**Contribution:** 1
**Rating:** 2
**Confidence:** 4

**Summary:**

This paper investigates whether VLMs can learn generalizable intuitive physics through interaction, as opposed to passive SFT. The authors compare two methods—GRPO (an interactive RL method) and SFT (non-interactive)—on tasks involving block tower stability and construction. They test three hypotheses: whether interactive training improves (1) within-task generalization, (2) cross-task generalization, and (3) sample efficiency on new tasks. The results show that while both methods achieve near-ceiling performance on trained tasks, neither leads to robust generalization to related tasks, suggesting that current fine-tuning methods encourage shortcut learning rather than genuine physical reasoning.

**Strengths:**

1. The paper is one of the first to systematically compare interactive (RL) and non-interactive (SFT) training for intuitive physics in VLMs.
2. The writing is clear, and the experimental setup is well-explained.

**Weaknesses:**

1. Limited Experimental Scope: The study is confined to a single model family (Qwen2.5-VL) and a narrow set of block-stacking tasks. Broader evaluation across diverse model architectures and more varied physical reasoning tasks would strengthen the conclusions.

2. Limited Innovation: The paper only compares SFT and RL on a small task without further analysis of the underlying reasons or proposing potential solutions—or at least offering improvements specific to the physics subtask. Additionally, it does not explore whether longer training, alternative RL algorithms, or more diverse interaction strategies could enhance generalization.

3. Limited Community Impact: The paper’s point about not overestimating the generalization ability brought by SFT (L459) appears to be a widely recognized observation. Moreover, the study focuses only on the narrow subfield of physical perception and is validated on a small-scale dataset, which limits the significance of its findings.

4. Presentation Issues:
(a) Appendix A.5 Attention Maps: It is unclear what the authors intend to illustrate with these visualizations.
(b) Related Work: The related work section is overly verbose and fails to clearly highlight the paper’s significant contributions compared to prior research.

**Questions:**

See weaknesses.
I am willing to chat with the authors to improve the paper.

---

> ### Author Response · Authors · 2025-11-21
> **Author response to Reviewer aHQU (1)**
>
> Dear Reviewer aHQU,
>
> Thank you for your review. We are happy to hear that you find that our “writing is clear, and the experimental setup is well-explained”. We especially appreciate your willingness to engage with us to improve the paper. In the following, we will address the individual points you raised.
> \
> \
> **Limited Experimental Scope: The study is confined to a single model family (Qwen2.5-VL) and a narrow set of block-stacking tasks.  Broader evaluation across diverse model architectures and more varied physical reasoning tasks would strengthen the conclusions.**
>
> Thank you for your comment. To improve the scope of our experiments and to test the generality of this result, we have added two further models (Qwen3-8B and Qwen3-32B) over the course of the rebuttal (see Section 4.5.2 and Appendix A.8.4) — both post-trained using SFT, GRPO and a newer RL variant called GSPO. Given time and resource constraints, we train these models on one task (x-only top block) and test how well they generalize to all other tasks. For these models, we find that they again perform well on the task they are trained on. However, we also find that they show some generalization to the related “binary-stability” task (see Figure 4 in the updated manuscript). These two tasks share the same data and the same properties of interest, with the offset of the top block being the relevant property to solve both tasks. However, we still find that they do not generalize to the other related tasks, such as for example “x-only side block”. This task is the same as the models’ post-training task, only with larger block displacements. If the models had learned the mapping between block displacement and the integer action space, they should in principle be able to solve this task as well — however, we find that they do not. Furthermore, we tested models on judging the stability of real images of block towers from [1]. We again find that no model performs well on this task, regardless of model age, model size, and if they are trained with interaction or not.
>
> To summarize, while we find some traces of generalization between the “x-only top block” and ”binary-stability” tasks in these newer models, it is still very limited and the models do not generalize to other related tasks. We hypothesized that interaction would be helpful for learning generalizable physical intuitions. However, we again do not find evidence for this: while the GRPO/GSPO 8B models achieve slightly higher accuracies on generalizing to the “binary-stability” task than the SFT 8B model, the 32B model trained with GRPO/GSPO does not transfer to this task at all, while the respective SFT model does. As such, we still do not find clear evidence that training these models with interaction gives them generalizable physical intuitions, nor that it is better than SFT when it comes to generalization to related tasks.
>
> Please note that we also tried to post-train Gemma3-12B and 32B, as well as Pixtral-12B. However, training these models with GRPO is not always straightforward. We attempted a number of hyperparameter combinations (top_p versus min_p sampling, different temperatures, different combinations of batch size and number of generations, and introducing KL regularization), however we were not able to successfully train either model on the “x-only top block” task over the course of the rebuttal. The Pixtral model struggled with basic formatting requirements, while the Gemma models got stuck in a local minimum, responding with 0 for all images (analogous to the random baseline we report).
>
> We have also added a number of additional ablations to ensure the robustness of our results (see section A.8) — we train the GRPO models for longer, we train models with lower ranks, we train models with reasoning, we train models without fine-tuning the vision encoder, we train models on multiple tasks jointly, we train models on a multi-step variant of the side block data set, and finally we test models’ generalization to real images.

---

> > ### Author Response · Authors · 2025-11-21
> > **Author response to Reviewer aHQU (2)**
> >
> > **Limited Innovation: The paper only compares SFT and RL on a small task without further analysis of the underlying reasons or proposing potential solutions—or at least offering improvements specific to the physics subtask. Additionally, it does not explore whether longer training, alternative RL algorithms, or more diverse interaction strategies could enhance generalization.**
> >
> > Thank you for these valuable suggestions. We have now run experiments to see if any of these suggestions can improve generalization (see section A.8 in the updated manuscript). In terms of model ablations, we train and evaluate models with lower LoRA ranks, with long-context (‘reasoning’) outputs, and without fine-tuning the vision encoder, as well as training the GRPO models for up to 50k steps. We also train models on multiple tasks jointly: we train both SFT and GRPO first on 10k steps of the x-only side tower task and then on for 10k steps on the binary stability top tower task. We also train an SFT model for 10k steps on a mixed data set of instances from both of these tasks. Finally, we train models with a new RL algorithm, Group-Sequence Policy Optimisation (GSPO; see section A.8.4 in the Appendix of the updated manuscript).
> >
> > We find that no training ablation produces models that generalize from their post-training task. For the models trained on multiple tasks jointly, we find that the GRPO models perform well at both tasks, while SFT models trained in a blocked manner (first one task, then another) seem to forget the first post-training task after some exposure to the second task. As for the alternative RL algorithm, we find that some models trained with GSPO show traces of generalization, which we discussed more in the paragraph above (you can find more details in section 4.5.2 and A.8.4 of the updated manuscript).
> > \
> > \
> > **Limited Community Impact: The paper’s point about not overestimating the generalization ability brought by SFT (L459) appears to be a widely recognized observation. Moreover, the study focuses only on the narrow subfield of physical perception and is validated on a small-scale dataset, which limits the significance of its findings.**
> >
> > We agree that previous work has already established that models post-trained with SFT do not generalize well [1, 2]. While we very much appreciate your assessment that our work is “one of the first to systematically compare interactive (RL) and non-interactive (SFT) training for intuitive physics”, we want to stress that comparing SFT and GRPO was not the main objective of our study.
> >
> > We hypothesized that reinforcement learning based methods would lead to models that generalize better, given that they allow models to interact with an environment, similar to how we think children learn about the physical world. In this framework, the SFT results act as a non-interactive baseline. However, we find that neither method results in models that generalize well, even to simple related tasks. This is in contrast to recent findings [1] and highlights that building VLMs with physical intuitions will likely require more than post-training on a restricted set of tasks.
> >
> > We wanted to take the chance to highlight that we specifically use this narrow set of block-stacking tasks as a well-designed and highly controlled test-bed to investigate the generalizability of core physical knowledge in VLMs. Physical reasoning is widely recognized as a necessary component for human-like machine learning models [3] and this controlled approach allows us to gain some understanding about the effects of the post-training methods we can use to improve it.

---

> > > ### Author Response · Authors · 2025-11-21
> > > **Author response to Reviewer aHQU (3)**
> > >
> > > **Presentation Issues: (a) Appendix A.5 Attention Maps: It is unclear what the authors intend to illustrate with these visualizations. (b) Related Work: The related work section is overly verbose and fails to clearly highlight the paper’s significant contributions compared to prior research.**
> > >
> > > Thank you for your thorough engagement with our analyses. We agree that the take-aways from the attention maps are not straightforward. Initially, we wanted to use the attention maps to better understand what strategies the post-trained models use, and to see if any pixel-level shortcuts emerge clearly. However, we found that it was hard to read anything from the attention maps at all. We only included them here to faithfully report every analysis we ran, including negative results.
> > >
> > > In order to highlight this more clearly, we rewrote the explanation in the Appendix of the updated manuscript. In the initial manuscript, we wrote “As seen in the example below, there is no clear change as a result of the post-training method that would give us a better understanding of the strategy used by the post-trained models.” — we changed this to say “As illustrated by the examples below, it is not possible to easily understand what the models attend to given the attention maps. We do not find clear changes as a result of either post-training method that would give us a better understanding of the strategy used by the post-trained models.”
> > >
> > > As for the related work section, we have rewritten it to be less verbose and to focus more on our paper’s contribution in relation to previous work (see section 2 in the updated manuscript).
> > > \
> > > \
> > > Thank you again for your time and for engaging so thoroughly with our work! We believe your suggestions have made our manuscript more robust and that based on them, we were able to improve the scope of our contributions.
> > > \
> > > \
> > > [1] Chu, Tianzhe, et al. "Sft memorizes, rl generalizes: A comparative study of foundation model post-training." arXiv preprint arXiv:2501.17161 (2025).\
> > > [2] Schulze Buschoff, Luca M, et al. "Testing the Limits of Fine-Tuning for Improving Visual Cognition in Vision Language Models." Forty-second International Conference on Machine Learning.\
> > > [3] Lake, Brenden M., et al. "Building machines that learn and think like people." Behavioral and brain sciences 40 (2017): e253.

---

### Official Review · Reviewer_Tj6t · 2025-10-30

**Soundness:** 3
**Presentation:** 3
**Contribution:** 3
**Rating:** 6
**Confidence:** 4

**Summary:**

The paper asks whether interaction helps VLMs acquire generalizable intuitive physics. Using Two TDW tower datasets and four tasks, the authors fine-tune Qwen2.5 VL with SFT or GRPO. Both reach near ceiling on the task they train on, yet show little transfer to related tasks. Linear probes can decode relevant physical quantities from activations, but this competence does not translate into zero-shot performance. Additional SFT on a new task learns faster than from the base model. Overall the study delivers careful negative results (which I appreciate a lot that the authors shared this.)

**Strengths:**

- Clear question and hypotheses grounded in cognitive science
- Controlled comparison of SFT and RL with matched PEFT settings and budgets
- Simple tasks with explicit rewards and prompt templates, plus training logs
- Generalization matrix across all train and test task pairs
- Decodability analysis that separates representation competence from output performance
- Useful visualization of reward landscapes and attention maps
- Negative results are reported transparently

**Weaknesses:**

- Very narrow scope. One model family at one size and one environment
- Interaction is minimal. One step RL with short textual actions, not true multi-step closed loop control
- Fixed camera and block sizes make pixel shortcuts likely, which undermines conclusions about physics learning
- No baselines for multitask SFT, joint training across tasks, or auxiliary representation losses
- Linear probe dataset is small and lacks controls such as image only probes or interventions

**Questions:**

- What happens with multi-step interaction and longer action horizons?
- Does heavy domain randomization of textures, lighting, camera pose, and block size improve transfer?
- How does joint multitask SFT across all four tasks compare to single task post-training?
- Do larger backbones or different families change the outcome?
- Can you test on a external stability dataset to validate generality?

---

> ### Author Response · Authors · 2025-11-21
> **Author response to Reviewer Tj6t (1)**
>
> Dear Reviewer Tj6t,
>
> Thank you for your thorough review. We are glad that you think our question and hypotheses are clear and “grounded in cognitive science”. Furthermore, we appreciate your assessment of our study as “controlled” with “useful visualizations”. In the following, we will try to address the weaknesses and questions that you have brought to our attention.
> \
> \
> **Very narrow scope. One model family at one size and one environment. Do larger backbones or different families change the outcome?**
>
> Thank you for raising this point. To improve the scope of our experiments and to test the generality of this result, we have added two further models (Qwen3-8B and Qwen3-32B) over the course of the rebuttal (see Section 4.5.2 and Appendix A.8.4) — both post-trained using SFT, GRPO and a newer RL variant called GSPO. Given time and resource constraints, we train these models on one task (x-only top block) and test how well they generalize to all other tasks. For these models, we find that they again perform well on the task they are trained on. However, we also find that they show some generalization to the related “binary-stability” task (see Figure 4 in the updated manuscript). These two tasks share the same data and the same properties of interest, with the offset of the top block being the relevant property to solve both tasks. However, we still find that they do not generalize to the other related tasks, such as for example “x-only side block”. This task is the same as the models’ post-training task, only with larger block displacements. If the models had learned the mapping between block displacement and the integer action space, they should in principle be able to solve this task as well — however, we find that they do not. Furthermore, we tested models on judging the stability of real images of block towers from [1]. We again find that no model performs well on this task, regardless of model age, model size, and if they are trained with interaction or not.
>
> To summarize, while we find some traces of generalization between the “x-only top block” and ”binary-stability” tasks in these newer models, it is still very limited and the models do not generalize to other related tasks. We hypothesized that interaction would be helpful for learning generalizable physical intuitions. However, we again do not find evidence for this: while the GRPO/GSPO 8B models achieve slightly higher accuracies on generalizing to the “binary-stability” task than the SFT 8B model, the 32B model trained with GRPO/GSPO does not transfer to this task at all, while the respective SFT model does. As such, we still do not find clear evidence that training these models with interaction gives them generalizable physical intuitions, nor that it is better than SFT when it comes to generalization to related tasks.
>
> We have also added a number of additional ablations to ensure the robustness of our results (see section A.8) — we train GRPO models for a longer time, we train models with lower ranks, we train models with reasoning, we train models without fine-tuning the vision encoder, we train models on multiple tasks jointly, we train models on a multi-step variant of the side block data set, and finally we test models’ generalization to real images.
> \
> \
> **Interaction is minimal. One step RL with short textual actions, not true multi-step closed loop control. What happens with multi-step interaction and longer action horizons?**
>
> Thank you for this suggestion. We agree that our implementation constitutes only a first step towards true interaction. In the RL literature, one-step learning tasks (i.e., bandits) are taken to be a special case of the more general n-step problem, all of which constitute some degree of interaction with an environment [2].
> While generalising our approach to multi-step interaction would be an ideal extension of our work, it is currently beyond what is computationally feasible. Using Group-Relative/-Sequence Policy Optimisation means that the model produces a batch of action sequences (token outputs) for a single state, which are then ranked by the reward signal. To produce an n-step sequence, the model’s actions need to be fed into unique instances of the environment to generate new states, so the number of active environment instances grows exponentially as the batch size to the power of the number of steps. For a 2-step interaction with a batch size of 16 needs 64 active instances of the physics engine. This is currently beyond our computational capacity. Future work will explore using cheaper, less realistic physics engines to improve compute overhead and training time.

---

> > ### Author Response · Authors · 2025-11-21
> > **Author response to Reviewer Tj6t (2)**
> >
> > **Fixed camera and block sizes make pixel shortcuts likely, which undermines conclusions about physics learning**
> >
> > We agree that it is possible that models learn shortcuts instead of proper physical reasoning. We purposefully fixed the camera angle and kept block sizes constant, so that models can learn the mapping between the input pixel space and the action space (understanding that moving an object “+100” relates to moving “X” pixels to the right). This should in theory allow models to transfer between tasks.
> > However, the failures we see in generalization for these models seem to point out that they do not learn to use this mapping. However, it is not immediately obvious what pixel level shortcut the x-only interaction models could learn. We attempted to use the attention maps to identify whether these models were attending to particular pixels during the task, but no clear pattern emerges.
> > \
> > \
> > **No baselines for multitask SFT, joint training across tasks, or auxiliary representation losses. How does joint multitask SFT across all four tasks compare to single task post-training?**
> >
> > Thank you for these excellent suggestions, we think training models jointly over tasks is a great idea and we have added a section about this in the updated version of our paper (see section 4.5.2 and A.8.3 in the updated manuscript).
> > To test if generalization can be improved by joint post-training on multiple tasks, we train a GRPO model first on “x-only side block” and then on “binary-stability top block”. Since this model has been trained on the x-only task and also on the top block data set (however not concurrently) we expected this model to generalize well to the “x-only top block” task.
> > We find that the GRPO model that has been trained on both tasks can still perform both tasks (see section A.8.3 in the Appendix of the updated manuscript). The model has some trouble keeping the correct formatting for the first task block it was trained on, but filtering only legal answers reveals that it still retains the capacity to solve it. However, the model still does not generalize to the “x-only top block” task, even though it has seen both the  “x-only” task and also the “top block” data set (albeit not at the same time).
> >
> > We also train an SFT model in the same blocked manner, first training it on “x-only side block” and then on “binary-stability top block”. The SFT model on the other hand quickly degrades in performance on the first of the blocked tasks. This indicates some benefit of GRPO when training models on multiple tasks successively.
> > We also test if interleaved joint training improves generalization by training an SFT variant with interleaved data, effectively training it on “x-only side block” and “binary-stability top block” at the same time. This model does not suffer from degraded performance in either task, reproducing results on joint-training from [3].

---

> ### Author Response · Authors · 2025-11-21
> **Author response to Reviewer Tj6t (3)**
>
> **Linear probe dataset is small and lacks controls such as image only probes or interventions**
>
> Indeed the size of the probing dataset is small. However, we believe that the high cross-validated accuracies achieved despite this constraint is an even stronger indicator of the required representations being present in the model. Given larger probing datasets it is likely that the probes would have achieved even higher accuracies.
> As for image-only probes, the linear probes are trained over the image token representations only. Because the image tokens appear before the text tokens, the probes only have access to the visual information. As such, the probes we train can be considered image-only. Thank you for providing us with the chance to clarify this. We have also expanded on this in the updated manuscript (see Section A.6 in the Appendix of the updated manuscript).
> \
> \
> **Does heavy domain randomization of textures, lighting, camera pose, and block size improve transfer? Can you test on a external stability dataset to validate generality?**
>
> Thank you for highlighting this. We have added test results for a data set showing pictures of real block towers [2]. Similar to previous work [3], we find that generalization from this type of task-specific post-training is so limited that even models trained on the same task (binary-stability top block) do not generalize from their artificial training set to this real test set (see also section A.8.5 in the Appendix of the updated manuscript).
>
> With respect to randomizing the data generated from ThreeDWorld, we would like to emphasise that camera pose and block size need to be constant so that models can learn the mapping between pixel and action space. We ensure all data sets share the same visual characteristics to give models the best chance to generalize. To ensure that the adapters for the models do not overfit, we test whether lower rank fine-tuning produces the same failures to generalize (see section A.8.2 in the Appendix of the updated manuscript). These models show the same failure to generalize, even though the chances of them overfitting are much lower. We also train a model without vision-finetuning to ensure our models do not overfit on the visual characteristics of the scene. This model again shows the same failure to generalize to other tasks (see Figure 14 in the Appendix).
> \
> \
> We would like to thank you once again for your review, which we think has improved our manuscript.
> \
> \
> [1] Lerer, Adam, Sam Gross, and Rob Fergus. "Learning physical intuition of block towers by example." International conference on machine learning. PMLR, 2016.\
> [2] Sutton, R. S., & Barto, A. G. (1998). Reinforcement learning: An introduction (Vol. 1, No. 1, pp. 9-11). Cambridge: MIT press.\
> [3] Schulze Buschoff, Luca M., et al. "Testing the Limits of Fine-Tuning for Improving Visual Cognition in Vision Language Models." Forty-second International Conference on Machine Learning.

---

### Official Review · Reviewer_rGrg · 2025-10-31

**Soundness:** 2
**Presentation:** 2
**Contribution:** 2
**Rating:** 2
**Confidence:** 4

**Summary:**

This paper explores whether supervised fine tuning works better than reinforcement learning fine tuning for vision language models in the context of intuitive physics tasks. They construct a train and test dataset using the ThreeDWorld simulator where structures are constructed with multiple blocks (i.e cubes), and an agent is required to either make a judgement of whether the structure is stable or how much a given block should be moved to make the structure stable. Vision language models are then trained on this task using supervised finetuning or reinforcement learning and tested to compare relative performance.

The authors find that there is not much difference in the performance of these two classes of models. They perform at ceiling when tested on the same kind of tasks seen in training and generalize equally poorly to new physical tasks.

**Strengths:**

1. Training models to understand intuitive physics by interacting with the environment  is a well motivated hypothesis, as it is similar to how babies learn.
2. The proposed metrics to test models seems reasonable, and the authors conduct rigorous evaluations.

**Weaknesses:**

1. The dataset seems a bit simple and contrived. The fact that supervised learning performs as good as reinforcement learning might be because it’s a really easy task, and not because both methods fundamentally work equally well.
2. I strongly disagree with the statement made in the conclusion that “these results cast doubt on whether posttraining methods are sufficient for developing models that reason about the world in a human-like manner”. The models not generalizing to new tasks, might just be because the training set is nowhere close to the amount of data that babies see, and not because the training algorithm is limited in some way. So we can’t really conclude anything about which model class is better from this result.
3. It seems like the rewards are really handcrafted for this particular task. This is certainly not how humans would get rewards in the real world, so I’m curious to know what the authors think about how this method would scale to multiple tasks in different environments. Would rewards need to be defined for each task separately?
4. It’s also not clear whether babies need a particular set of task specifications and goals for being able to learn intuitive physics. Most of intuitive physics might be learnt just by passive object manipulations without any defined goal like stability or placement. So how would we control for this kind of variable in the experiment? It makes me think that there is an inherent limitation in the way the training pipeline is set up here, which again makes me less confident about making any conclusions.

**Questions:**

My main concerns are about the dataset being too simple. How could this be extended or improved to ensure that the conclusions are reliable?

---

> ### Author Response · Authors · 2025-11-21
> **Author response to Reviewer rGrg (1)**
>
> Dear Reviewer rGrg,
>
> Thank you for your review. We are happy to hear that you thought our hypothesis was “well motivated” and that you find that we “conduct rigorous evaluations”. In the following, we will try to address the questions you raised.
> \
> \
> **The dataset seems a bit simple and contrived. The fact that supervised learning performs as good as reinforcement learning might be because it’s a really easy task, and not because both methods fundamentally work equally well.**
>
> Thank you for raising this point — we agree that the tasks we use are simple. Inspired by previous work in developmental psychology [1] and machine learning [2, 3] which use similar tasks to test physical reasoning, we specifically chose tasks that vary only along specific dimensions to pinpoint to what degree models are able to generalize from their post-training task to related tasks with similar physical dynamics and visual characteristics. Since the tower-building and stability judgment tasks come with simple dynamics and ground truth data, they serve as a good test-bed.
>
> Physical reasoning is widely recognized as a necessary component for human-like machine learning models [4] and this controlled approach allows us to gain some understanding about the generalizability of core, common-sense knowledge in VLMs. Our point therefore is not to argue that SFT and GRPO work equally well on these simple tasks. We hypothesized that reinforcement learning based methods would lead to models that generalize better, given that they allow models to interact with an environment, similar to how we think children learn about the physical world — in this framework, the SFT results are supposed to act as a non-interactive baseline. However, we find that neither method results in models that generalize well, even to these simple related tasks. This is in contrast to recent findings [5] and highlights that building VLMs with physical intuitions will likely require more than post-training on a restricted set of tasks.
> \
> \
> **I strongly disagree with the statement made in the conclusion that “these results cast doubt on whether posttraining methods are sufficient for developing models that reason about the world in a human-like manner”. The models not generalizing to new tasks, might just be because the training set is nowhere close to the amount of data that babies see, and not because the training algorithm is limited in some way. So we can’t really conclude anything about which model class is better from this result.**
>
> Thank you for raising this point. While it is possible that these models have seen less training data than babies, we would contend that these models are pre-trained on vast amounts of visual and textual data, a significant fraction of which will include images and descriptions of physical phenomena. The developers of Qwen-2.5 report that their models were trained on 4 trillion tokens, including “a wide variety of multimodal data, such as image captions, interleaved image-text data, optical character recognition (OCR) data, visual knowledge (e.g., celebrity, landmark, flora, and fauna identification), multi-modal academic questions, localization data, document parsing data, video descriptions, video localization, and agent-based interaction data” [6].
> According to some hypotheses, fine-tuning allows models to appropriately apply the knowledge they learnt through pretraining [7]. We would therefore argue that our approach of taking pre-trained models that have seen trillions of tokens during training, and then having them learn and apply that knowledge through interaction, is not entirely unlike children' s learning. Children also learn through interaction with their environment, applying knowledge they have acquired before and refining their intuition through seeing what effects their actions have on this environment.
>
> We have described this reasoning further in the text. We have also softened our claim by highlighting the task-specific nature of our post-training, it now reads: “It remains unclear whether post-training models on specific cognitive tasks is sufficient for developing models that reason about the world in a human-like manner.”

---

> ### Author Response · Authors · 2025-11-21
> **Author response to Reviewer rGrg (2)**
>
> **It seems like the rewards are really handcrafted for this particular task. This is certainly not how humans would get rewards in the real world, so I’m curious to know what the authors think about how this method would scale to multiple tasks in different environments. Would rewards need to be defined for each task separately?**
>
> Thank you for raising this question. Our reward function was hand-crafted to match the tower building tasks we designed, similarly to how reward functions are designed for game-playing agents or for agents tasked with generating code that passes unit tests. You are correct to point out that there is significant room to generalise our reward functions for more general physical tasks, but it is nonetheless the case that any general agent trained with reinforcement learning will do so on the basis of one or several human-designed reward functions.
>
> While it is the case that there are certain extrinsic reward signals that are not designed by humans per se – but rather by evolutionary dynamics, such as food-seeking – intrinsic reward signals can originate from individuals or human groups. For instance, a Piagetian might argue that building a higher block tower is intrinsically and/or socially rewarding for a child, and that such reward signals are a key component of play behaviour in many animal species [8-10]. Therefore, there is an argument to be made that some human reward signals are crafted by humans too. However, we would also like to stress that we do not claim to be trying to replicate human learning in machines, but rather, we use the human case as inspiration for a novel approach to training VLMs that are better at intuitive physics.
>
> Thank you for your suggestion with training models on multiple tasks in different environments. We think it is very interesting to see if models can learn tasks with different rewards functions and in different environments simultaneously. We have therefore trained a GRPO model using a blocked approach where it is first trained on the “x-only” task for the “side block” data set, and then on the “binary-stability” task on the “top block” data set.
>
> Since this model has been trained on the x-only task and also on the top block data set (however not concurrently) we expected this model to generalize well to the “x-only top block” task. We find that the GRPO model that has been trained on both tasks can also perform both tasks (see section A.7.4 in the Appendix of the updated manuscript). The model has some trouble keeping the correct formatting for the first task block it was trained on, but filtering only legal answers reveals that it still retains the capacity to solve it. However, the model still does not generalize to the “x-only top block” task, even though it has seen both the  “x-only” task and also the “top block” data set (albeit not at the same time).
>
> Please note also that the “x-only” task is trained and evaluated using the exact same reward function for the “top block” and “side block” datasets. This means that models should be able to generalize between the two. However, our results show that they do not, highlighting that their failure to generalize is not only due to the specification of the reward function.

---

> ### Author Response · Authors · 2025-11-21
> **Author response to Reviewer rGrg (3)**
>
> **It’s also not clear whether babies need a particular set of task specifications and goals for being able to learn intuitive physics. Most of intuitive physics might be learnt just by passive object manipulations without any defined goal like stability or placement. So how would we control for this kind of variable in the experiment? It makes me think that there is an inherent limitation in the way the training pipeline is set up here, which again makes me less confident about making any conclusions.**
>
> We agree that children do not solely learn by way of extrinsic/intrinsic reward signals and supervision, but also through unsupervised strategies. However, significant research in developmental psychology suggests that infants learn much about their world through trial-and-error with it [10, 11]. Indeed, some have argued that play behaviour is reinforced by intrinsic reward signals - by an intrinsic reward that comes from understanding the world through interaction [12, 13].
>
> We should emphasise that physical understanding is something that develops over the first years of life, rather than the first months [14]. Importantly, this research inspires a way of thinking about how we might improve existing models with existing technology. It is not clear yet how we might train vision-language models with true human-like curricula, combining reinforcement, supervised, and unsupervised learning. However, we are able to attempt the first two, in isolation and in combination, given current language modelling techniques.
>
> We appreciate you raising this point and highlighting where we have not been clear enough in outlining our motivation. We have re-written parts of the introduction in our updated manuscript to better reflect this.
> \
> \
> We would like to thank you once again for your comments, which we think have improved the clarity of our investigation.
> \
> \
> [1] Baillargeon, Renée, Amy Needham, and Julie DeVos. "The development of young infants' intuitions about support." Early development and parenting 1.2 (1992): 69-78.\
> [2] Lerer, Adam, Sam Gross, and Rob Fergus. "Learning physical intuition of block towers by example." International conference on machine learning. PMLR, 2016.\
> [3] Battaglia, Peter, et al. "Interaction networks for learning about objects, relations and physics." Advances in neural information processing systems 29 (2016).\
> [4] Lake, Brenden M., et al. "Building machines that learn and think like people." Behavioral and brain sciences 40 (2017): e253.\
> [5] Chu, Tianzhe, et al. "Sft memorizes, rl generalizes: A comparative study of foundation model post-training." arXiv preprint arXiv:2501.17161 (2025).\
> [6] Bai, Shuai, et al. "Qwen2. 5-vl technical report." arXiv preprint arXiv:2502.13923 (2025).
> [7] Raghavendra, M., Nath, V., & Hendryx, S. (2024). Revisiting the superficial alignment hypothesis. arXiv preprint arXiv:2410.03717.\
> [8] Smith, P. K. (1982). Does play matter? Functional and evolutionary aspects of animal and human play. Behavioral and brain sciences, 5(1), 139-155.\
> [9] Chu, J., & Schulz, L. E. (2020). Play, curiosity, and cognition. Annual Review of Developmental Psychology, 2(1), 317-343.\
> [10] Nicolopoulou, A. (1993). Play, cognitive development, and the social world: Piaget, Vygotsky, and beyond. Human development, 36(1), 1-23.\
> [11] Smith, L., & Gasser, M. (2005). The development of embodied cognition: Six lessons from babies. Artificial life, 11(1-2), 13-29.\
> [12] Gopnik, A., Meltzoff, A. N., & Kuhl, P. K. (1999). The scientist in the crib: Minds, brains, and how children learn. William Morrow & Co.\
> [13] Schulz, L. E., & Bonawitz, E. B. (2007). Serious fun: preschoolers engage in more exploratory play when evidence is confounded. Developmental psychology, 43(4), 1045.\
> [14] Baillargeon, R., Li, J., Gertner, Y., & Wu, D. (2010). How do infants reason about physical events?. The Wiley‐Blackwell handbook of childhood cognitive development, 11-48.

---

### Official Review · Reviewer_aSaP · 2025-11-01

**Soundness:** 2
**Presentation:** 2
**Contribution:** 2
**Rating:** 2
**Confidence:** 3

**Summary:**

This paper investigates whether VLMs better learns physical reasoning through SFT versus RL (GRPO).
They use a custom block tower building benchmark with two tasks:
- binary stability prediction
- block displacement prediction for tower stabilization

They compare two training approaches, using the Qwen2.5-VL 7B model with PEFT:
(1) SFT: Supervised fine-tuning on labeled correct answers
(2) GRPO: RL where the model predicts displacements/stability and receives rewards based on whether predictions would result in stable towers

**Strengths:**

The research question is well-motivated - comparing RL versus passive supervised learning for physical reasoning is an important question for embodied VLAs and robotics.

The method section is clearly written and self-contained, with sufficient implementation details.

The experimental design is clean and controlled, the writing of the experimental section is ordered nicely,
as each experiment motivates the following ones.

**Weaknesses:**

While this serves as a useful motivating example, the scope of these experiments (single model, relatively narrow custom task) makes it difficult to draw significant conclusions, and it's unclear whether these findings can be generalized to larger scale training with multiple tasks.
I like the direction of this work but I believe the findings and experiments are too limited for a full publication at ICLR.

The results are hard to parse - a simple table summarizing performance and rewards would complement Figure 2 well.

**Questions:**

What we can actually take away from this work for larger scale training of spatial / physics understanding for VLMs?
What is the message the authors are trying to convey with this work?

The VLAs in robotics are trained on physically grounded tasks (pick and place, manipulation) and might learn some of this "physics concepts" implicitly through their training.
I wonder if this actually makes a difference compared to the data and task Qwen2.5-VL is trained on.

---

> ### Author Response · Authors · 2025-11-21
> **Author response to Reviewer aSaP (1)**
>
> Dear Reviewer aSaP,
>
> Thank you for your thoughtful review. We appreciate your assessment of our paper as “well-motivated” and “clearly written” and that you think our “experimental design is clean and controlled”. In the following, we will address the individual points you have raised.
> \
> \
> **While this serves as a useful motivating example, the scope of these experiments (single model, relatively narrow custom task) makes it difficult to draw significant conclusions.**
>
> Thank you for highlighting this. To improve the scope of our experiments and to test the generality of this result, we have added two further models (Qwen3-8B and Qwen3-32B) over the course of the rebuttal (see Section 4.5.2 and Appendix A.8.4) — both post-trained using SFT, GRPO and a newer RL variant called GSPO. Given time and resource constraints, we train these models on one task (x-only top block) and test how well they generalize to all other tasks. For these models, we find that they again perform well on the task they are trained on. However, we also find that they show some generalization to the related “binary-stability” task (see Figure 4 in the updated manuscript). These two tasks share the same data and the same properties of interest, with the offset of the top block being the relevant property to solve both tasks. However, we still find that they do not generalize to the other related tasks, such as for example “x-only side block”. This task is the same as the models’ post-training task, only with larger block displacements. If the models had learned the mapping between block displacement and the integer action space, they should in principle be able to solve this task as well — however, we find that they do not. Furthermore, we tested models on judging the stability of real images of block towers from [1]. We again find that no model performs well on this task, regardless of model age, model size, and if they are trained with interaction or not.
>
> To summarize, while we find some traces of generalization between the “x-only top block” and ”binary-stability” tasks in these newer models, it is still very limited and the models do not generalize to other related tasks. We hypothesized that interaction would be helpful for learning generalizable physical intuitions. However, we again do not find evidence for this: while the GRPO/GSPO 8B models achieve slightly higher accuracies on generalizing to the “binary-stability” task than the SFT 8B model, the 32B model trained with GRPO/GSPO does not transfer to this task at all, while the respective SFT model does. As such, we still do not find clear evidence that training these models with interaction gives them generalizable physical intuitions, nor that it is better than SFT when it comes to generalization to related tasks.
>
> We agree that the tasks we train and evaluate the models on are narrow. We are using these physical reasoning tasks as a test-bed to investigate generalization in core abilities of machine learning models. Physical reasoning is widely thought of as a core component for human-like machine learning models [2]. Since the tower building and stability judgement tasks come with simple dynamics and definable ground truth, they serve as an ideal test-bed to investigate generalization in the physical reasoning domain. The binary stability task specifically has a long history in developmental psychology [3], as well as a history of being used as a test of physical reasoning in machine learning models [1, 4]. Inspired by these studies, we train and evaluate models on this relatively narrow set of tasks, which only differ along specific dimensions, so that we can pinpoint to what degree models are able to generalise in a controlled setting. We show that, even given these similar tasks which require models to learn about the same variables of interest, such as block displacement, Qwen2.5-VL-7B models post-trained with SFT or GRPO do not generalize reliably.
>
> To ensure the robustness of our results, we also added a number of additional ablations during the rebuttal (see section A.8 in the updated manuscript) — we train the GRPO models for longer, we train models with lower ranks, we train models with reasoning, we train models without fine-tuning the vision encoder, we train models on multiple tasks jointly, and as mentioned above, we test models’ generalization to real images.
> \
> \
> **The results are hard to parse - a simple table summarizing performance and rewards would complement Figure 2 well.**
>
> Thank you for suggesting this. We agree that Figure 2 can be hard to parse quickly as it shows within-task performance and generalization performance at the same time. We have added a table summarizing the performance of the fully post-trained GRPO and SFT models to the Appendix (see Section A.2).

---

> > ### Author Response · Authors · 2025-11-21
> > **Author response to Reviewer aSaP (2)**
> >
> > **What can we actually take away from this work for larger scale training of spatial / physics understanding for VLMs?**
> >
> > Thank you for raising this point. While our results focus on relatively small models on specific physical reasoning tasks, they show that the models we investigate have a hard time learning generalisable physical intuitions from any type of post-training. We find that models have a hard time generalising to even very related and similar tasks.
> >
> > To get a bit closer to what would happen if you post-trained models on large scale physical reasoning data sets, we trained models on two tasks jointly, to see if this confers better generalisation. Specifically, this model is first trained on the “x-only” task for the “side block” data set, and then on the “binary-stability” task on the “top block” data set. Since this model has been trained on the “x-only” task and also on the “top block” data set (however not concurrently) we expected this model to generalize well to the “x-only top block” task.
> >
> > We find that the GRPO model that has been trained on both tasks can also perform both post-training tasks (see section A.8.3 in the Appendix of the updated manuscript). However, the model still does not generalize to the “x-only top block” task, even though it has seen both the  “x-only” task and also the “top block” data set (albeit not at the same time). This highlights again how limited generalization from this type and scale of post-training is and might indicate that post-training on multiple tasks also does not lead to generalizable physical intuitions.
> > \
> > \
> > **The VLAs in robotics are trained on physically grounded tasks (pick and place, manipulation) and might learn some of these "physics concepts" implicitly through their training. I wonder if this actually makes a difference compared to the data and task Qwen2.5-VL is trained on.**
> >
> > We agree that Vision-Language-Action agents may learn some of these physical concepts implicitly through interacting with the environment. Indeed, this was our starting hypothesis, only we consider Vision-Language Models that produce actions given an image of the environment to be a useful proxy for VLA agents. This is because systematically querying pre-trained VLA agents is not practically possible. Our results are therefore limited to VLMs, and serve only to cast some doubt on whether VLAs can learn robust physical intuitions from interaction alone.
> > \
> > \
> > We would like to thank you once again for your comments, which we think have improved the scope and interpretation of our investigation.
> > \
> > \
> > [1] Lerer, Adam, Sam Gross, and Rob Fergus. "Learning physical intuition of block towers by example." International conference on machine learning. PMLR, 2016.\
> > [2] Lake, Brenden M., et al. "Building machines that learn and think like people." Behavioral and brain sciences 40 (2017): e253.\
> > [3] Baillargeon, Renée, Amy Needham, and Julie DeVos. "The development of young infants' intuitions about support." Early development and parenting 1.2 (1992): 69-78.\
> > [4] Battaglia, Peter, et al. "Interaction networks for learning about objects, relations and physics." Advances in neural information processing systems 29 (2016).

---

### Author Response · Authors · 2025-11-21
**Author general response**

Dear Reviewers and Area Chair,

We appreciate the reviewers taking time to engage with our work and want to thank them for their active participation in the review process. We were happy to hear that the reviewers found our paper “well-motivated” (aSaP, rGrg) and “clearly written” (aSaP, aHQU) and our experimental design “clean and controlled” (aSaP, rGrg, Tj6t).
\
\
In our individual responses to each reviewer, we tried to address their questions and concerns. Over the past week of the rebuttal, we have added the following:

- Newer and bigger models (aSaP, aHQU)
- Models trained on multiple tasks (rGrg, Tj6t, aHQU)
- Models trained with another RL algorithm, GSPO (aHQU)
- Evaluation on a related external dataset (aSaP, Tj6t, aHQU)
- Ablations for fine-tuning with reasoning, lower rank, without vision fine-tuning, and long-horizon training up to 50k steps (aHQU)
- Clarification of the related work section and discussion (aSaP, rGrg, Tj6t, aHQU)

\
We again want to thank our reviewers for their time and their productive comments, which we are confident have improved the scope and clarity of our results.

---

### Meta-Review · Area_Chair_xRXK · 2026-01-07

**Summary:**

All reviewers find the research question timely and the experimental setup clean and well controlled, and several appreciate the authors’ transparency in reporting a negative result. However, reviewers consistently raised concerns that the current evidence is too limited to support the paper’s broader conclusions about whether interaction helps vision–language models acquire generalizable intuitive physics. The main factors driving the decision are the narrow evaluation scope, with conclusions drawn largely from a block-tower benchmark and a restricted set of tasks; the limited form of interaction, which is largely short-horizon and does not reflect richer closed-loop physical interaction; and the reliance on task-specific reward design and fixed visual configurations, which raises the possibility of shortcut learning and weakens claims about physical reasoning. Reviewers also noted that the paper does not yet provide a clear actionable takeaway for the community beyond demonstrating a failure to generalize. The rebuttal adds meaningful experiments and clarifications, including larger models, an additional reinforcement learning variant, multitask and ablation studies, and evaluation on an external image dataset, but these additions do not fully resolve the core concerns.

Based on this assessment. The final recommendation is Reject.

**Reviewer Concerns:**

Across reviewers, the main concerns were narrow scope and limited interaction: `aSaP`, `rGrg`, and `aHQU` viewed the initial evidence as too limited to support broad conclusions, while `Tj6t` was more positive but still flagged minimal one-step interaction, shortcut risks, and missing multitask baselines. The rebuttal substantively addresses scope/presentation issues (`aSaP`/`aHQU`/`Tj6t`) by adding Qwen3-8B/32B, GSPO, longer/ablative training variants, sequential and interleaved multitask training, clearer summaries, and an external real-image stability test. It also softens over-strong claims (`rGrg`). However, the core limitations largely remain: interaction is still not meaningfully multi-step/closed-loop (`Tj6t`), reward design remains task-specific and the setting remains a constrained/contrived benchmark (`rGrg`, `aSaP`), and reviewers (`aHQU`, `aSaP`) still lack a compelling “what should the community do next?” takeaway beyond the negative result.

**Reviewer Scores:**

Given the rebuttal, I expect `aSaP` stays at 2 (improved breadth and readability but still narrow/generalization takeaway unchanged), `rGrg` stays at 2 (primary objection about drawing broader conclusions from limited-scale contrived tasks persists), `aHQU` likely stays at 2 or at most moves to 3 (added ablations help but innovation/impact concerns remain), and `Tj6t` likely stays around 6 since several requested experiments were added, but the minimal-interaction limitation still caps enthusiasm.

---

### Decision · Program_Chairs · 2026-01-26

Reject